# The Doubly Linked Tree of Singly Linked Rings: Providing Hard Real-Time Database Operations on an FPGA

Simon Lohmann *[ID] and Dietmar Tutsch *[ID]

Chair of Automation/Computer Science, University of Wuppertal, Rainer-Gruenter-Str. 21,
42119 Wuppertal, Germany
* Correspondence: slohmann@uni-wuppertal.de (S.L.); tutsch@uni-wuppertal.de (D.T.)

**Abstract:** We present a hardware data structure specifically designed for FPGAs that enables the execution of the hard real-time database CRUD operations using a hybrid data structure that combines trees and rings. While the number of rows and columns has to be limited for hard real-time execution, the actual content can be of any size. Our structure restricts full navigational freedom to every but the leaf layer, thus keeping the memory overhead for the data stored in the leaves low. Although its nodes differ in function, all have exactly the same size and structure, reducing the number of cascaded decisions required in the database operations. This enables fast and efficient hardware implementation on FPGAs. In addition to the usual comparison with known data structures, we also analyze the tradeoff between the memory consumption of our approach and a simplified version that is doubly linked in all layers.

**Keywords:** FPGA; RTDB; database; data structure; real-time; hard real-time; CRUD; ring; tree





## 1. Introduction

### 1.1. Motivation

#### 1.1.1. Hard Real-Time Databases

Classical database systems are not designed for real-time operation. The use of database concepts can, however, simplify the complicated task of designing real-time systems for tasks based on table data, e.g., tracing individual work pieces through automated production facilities to adapt further machining steps to the production history of the individual work pieces. Current real-time database (RTDB) systems with guaranteed deadlines work by gracefully aborting queries when a deadline cannot be met. While this may be a valid approach for some practical applications, it basically replaces the guarantee for a complete execution of the query with a guarantee for a timely answer, which is now allowed to be *deadline could not be met, query aborted*. Our long-time goal is to build a hard real-time RTDB that guarantees both the successful execution of the queries and their timely completion.

#### 1.1.2. Operations on Tables

Regarding the operations *Insert* and (*Create* in the commonly used *Create, Read, Update, Delete (CRUD)* acronym), *Update* and *Delete* build the foundation of table storage in a database system. While insertion at *any* position would be sufficient for a real-time database, inserting at *specific* positions can be beneficial for performance as it additionally allows storing orderedtables. (While relational databases do not have an inherent order of table entries, this can be useful, e.g., for time series). Apart from changing the dataset, we also need *Primary-Key-Access* to a specific row/value from its primary key and an operation to *Read* the actual data. Being able to execute all these operations in hard real-time is therefore essential.

*1.2. Issues to Address*

1.2.1. DB-Operations vs. Hard Real-Time

The CRUD operations operate on previously unknown amounts of data. As this is the first point where the data enter our system (or are deleted), we cannot assume that tricks like simply linking an existing object containing all the data in place via a pointer are applicable. We actually have to keep our system capable of hard real-time operation while touching every bit of information during Read/Insert/Update.

1.2.2. Dynamic Data Width

While most database columns in practical applications are probably of constant width (e.g., numbers, booleans, and timestamps), variable widths are common as well, e.g., for storage of strings or large numbers or binary large objects (BLOBs). This rules out purely static data structures like arrays.

1.2.3. Memory Allocation and Deallocation

Dynamic insertion and deletion of data requires the allocation and deallocation of memory. Updating a DB entry of dynamic width will most likely involve a change in the amount of used memory as well. As the allocated memory can shrink by large amounts with the deletion of a single row, an algorithm handling hard real-time deallocation of arbitrary amounts of memory is crucial. Allocation is equally important, but less complicated, since it can be completed step by step as new data arrive from a data stream (see Section 3.3).

1.2.4. Performance on FPGAs

As the implementations regarding our current research towards hard real-time databases take place on field programmable gate arrays (FPGAs), we also want to consider that FPGA designs favor parallel operations and/or pipelining. Long sequences of decisions, in contrast, come at the cost of reduced maximum operation frequency of the FPGA design. Keeping the amount of *dependent decisions* low for each clock cycle allows higher operation speeds. Our target is therefore to keep the amount of dependent decisions low, while the amount of parallel execution paths is not of concern.

*1.3. Contribution*

We present the *doubly linked tree of singly linked rings*, a hierarchical-node-based data structure providing the CRUD operations for hard real-time databases while keeping all elements in all hierarchy levels of identical structure, reducing the amount of sequential decisions (and completely omitting primary key lookup) and thus improving achievable performance on FPGAs. Only the typically small higher-hierarchy layers provide full navigation freedom (up/down/successor/predecessor), while the leaves are restricted to up/successor, reducing their memory footprint. It additionally provides interesting non-real-time options like hierarchic ascent and cyclic read from any entry point. We discuss the tradeoff vs. a completely doubly linked version and show in which aspects our structure outperforms well-known alternatives. Apart from an example interface, we also show how limited depth recursions used in one of our operations can be implemented for FPGAs in a synthesizable way using what we call a *context switching state machine*.

*1.4. Content*

The first section motivates the problem and presents issues to address. An overview of related work is provided in Section 2. In Section 3, we introduce our system model and explain basic notation and nomenclature. Section 4 presents our data structure and defines operations on it, followed by the corresponding real-time proofs in Section 5, an experimental evaluation in Section 6, and a comparison to other data structures as well as a double-linked ring variation in Section 7. Section 8 concludes the paper. An example interface is presented in Appendix A. Instructions on synthesizable implementation of the utilized limited depth recursion are provided in Appendix B.

## 2. Related Work

### 2.1. Real-Time Databases

A real-time database (RTDB) can process queries in real time. Typical applications include cyberphysical systems and stock trading. For hard real-time, every single query has to be finished before its individual deadline. Therefore, such systems must be designed with worst-case execution time (WCET) in mind—in contrast to non-real-time database systems, which are often optimized towards average case performance.

In practical applications, this restriction can be reduced to performing only queries involving database *content* in real time, while operations on the databases *schema* usually involve human interaction at some point (non-real-time). Another reason why we do not consider *schema* changes as hard real-time operations is that such a change implies reevaluation of the system's query execution plans and deadlines, which might find that the updated system cannot comply with the old deadlines. This is therefore better completed outside the hard real-time domain until deadline-compliant operations have been ensured by a real-time analysis.

**Remark 1.** *Classical database systems are not designed for real time. Despite some large players in the database market using the term "real-time" prominently in their advertising, what they sell would mostly be scientifically classified as active databases.*

Kao and Garcia-Molina [1] Ramamritham et al. [2,3], and Shanker et al. [4] provide good overviews on earlier publications in the field of RTDB research. A great deal of research has been completed on concurrency control [5] and transaction scheduling [6–9] and its influence on deadlines and new real-time-capable algorithms [10]. Other well-covered topics include locking protocols [11–16] and distributed real-time databases [5,13,17]. Only a few of the existing RTDB prototypes target *hard real-time* operation:

*MDARTS* by Lortz et al. [18] "provides hard real-time transaction times" by precalculating the worst-case execution time of every transaction. The authors discuss bounds on transaction overheads from preemption, locks, and interrupts. However, the quite essential calculation of actual transfer WCETs is not further explained; in the described experiments, WCETs are always "estimated" and "assumed" without further calculations. One could compare MDARTS to a real-time operating system: it provides a solid foundation on which the actual task (the real-time query) still has to be built.

Nyström et al. suggests *database pointers*, a mechanism for bypassing slow access paths in a DBMS [19], also targeting hard real-time control systems. The idea is to supplement a regular database (using regular index structures) with a second access layer, which is meant to be used for hot data (data queried at high frequency compared to the rest of the database). While the access method is more direct than the regular key lookup, it still provides features like type checking and locking via an additional *database pointer table*. Using a database pointer requires an initial call to the *bind(ptr, q)* operation, which looks up the physical address of the data in *q* and binds it to a new entry in the database pointer table.

*RTDB* by Nogiec and Desavouret [20] is "a fast, memory resident object database" with triggers, versioning, and bitfield-based attribute queries. It relies on the users directly using pointers after an initial name lookup, which provide direct access to the objects stored as is. The authors, however, do not elaborate on execution times, apart from a general statement that they see in-memory databases as "especially suited" for real-time systems due to their "short and predictable access times".

Another in-memory RTDB is the open source *KogMo-RTDB*, a time series object database by Goebl and Farber [21] that specifically targets event processing for cognitive automobiles. The history of each sensor is stored in a dedicated ring buffer of limited size [22]. KogMo-RTDB allows to query objects relative to a given timestamp (younger, older, or exact timestamp). It also provides queries that will wait for creation/deletion of objects. Another option is to query the most current data of an object, optionally waiting for the creation of new data if there are no data existing after the associated timestamp.

Introducing new object types is possible but requires recompilation of the database system. This includes compound object types with members of an existing type.

The recently released *eXtremeDB/rt* by McObject claims to be the "first" and "only" RTDB "that guarantees transaction deadlines" [23]. It achieves hard real-time aware execution of database queries by applying a policy of "successfully aborting" queries in hard real-time if it is detected that the deadline of a query cannot be met [24].

### 2.2. FPGAs and Databases

Field Programmable Gate Arrays (FPGAs) are configurable integrated circuits that host user-defined logic circuits. FPGAs are increasingly used in the database field [25,26]. Their applications are mostly limited to acceleration of database queries [27–30] controlled by a processor-based database management system, where they achieve quite remarkable speedups compared to CPU implementations. As FPGAs are reconfigurable, custom acceleration processors can be specifically adapted to the most intensive processing tasks.

*Glacier* [31,32] by Müller et al. compiles queries to custom FPGA designs, which are then synthesized for use in the target FPGA. This approach of synthesizing a dedicated full FPGA configuration for every new query is, however, only viable for very time-intensive workloads as synthesis of FPGA design itself is a rather time-consuming process and may easily eat up the expected speedup of smaller queries.

A different approach is the use of prebuilt processing modules in conjunction with partial reconfiguration of the FPGA. Queries are built by loading the correct modules into a universal datapath. This approach is used by Sukhwani et al. in [33,34], Dennl et al. in [35], and Ziener et al. in [28].

### 2.3. Hard Real-Time and Hardware Data Structures

Data structures for hard real-time mostly include well-known concepts like arrays (constant-time-index-based access) or linked lists (constant time insert/delete). Balanced tree structures like the B-tree [36] and its variants are also sometimes considered due to their low complexity of $\mathcal{O}(\log(N))$. As the depth of balanced trees grows with increasing data load, they are only capable of hard real-time when the amount of data is limited.

Hash tables are famous for their exceptional average performance of $\mathcal{O}(1)$, but they exhibit large execution times in worst-case scenarios. While the title of *Space Efficient Hash Tables with worst-case Constant Access Time* by Fotakis et al. [37] suggests a hash table with hard real-time capabilities, the presented method still suffers from the collision problem typical for hash tables: they "prove that at most $\gamma n$ vertices overflow whp" (with high probability, so not guaranteed), and moves on, only storing data without collisions, leaving the problematic cases to the usual collision-handling systems.

Bloom et al. suggests augmenting real-time systems with hardware data structures (HWDS) of commonly used data structures to speed up operations and reduce latency and jitter introduced by main memory accesses, resulting in overall WCET reduction [38]. Variations in the concept have been discussed earlier regarding specific target applications like network queues (Moon et al. [39]), sort queues (Kohutka and Stopjakova [40,41]), schedulers (Burleson et al. [42]) or string matching (Cameron and Lin [43]).

### 2.4. Dynamic Memory Management in FPGAs

Dynamic memory management is rarely seen in the world of FPGA designs. Most applications use FPGA-internal memory like LUT-RAM and Block-RAM, which is normally a direct part of the FPGA design architecture and fixed to the modules using it. Larger memory may be attached to the FPGA externally. However, even then, most FPGA applications assign fixed memory ranges to the individual modules in the FPGA design; this is also the approach in the AXI ecosystem by ARM [44]. This has the advantage that inter-client-communication can be performed by simply reading/writing the other client's memory range.

Although uncommon, approaches at (dynamic) memory management in FPGAs do exist: In *A Comprehensive Memory Management Framework for CPU-FPGA Heterogenous SoCs* [45], Du et al. presents a memory management framework for the Zynq-7000 MPSoC Series. This framework is targeted at use with the high level synthesis toolchain of Xilinx Vivado and seeks to automatically optimize the performance in designs with multiple clients by providing optimal data placement across BRAM and DDR-RAM as well as optimized cache partitioning based on the expected access frequency of the individual memory ranges assigned to the clients.

*Adaptive Dynamic On-Chip Memory Management for FPGA-based Reconfigurable Architectures* [46] by Dessouky et al. presents *DOMMU*, a dynamic memory management unit for sharing Block-RAM resources in an FPGA between multiple clients. It keeps a map of the currently unallocated, allocated, and deallocated BRAM elements. DOMMU includes a mechanism for automatic deallocation: If a BRAM is not accessed for a user-configurable time, its elements are automatically deallocated (except for the last element, which has to be deallocated manually) to make the memory available to other clients. The system is not targeted at hard real-time.

Another example of dynamic memory management on an FPGA is described in *FPGA Implementation of Memory Management for Multigigabit Traffic Monitoring* [47] by Trzepinski et al.: While this is an application-specific management approach for a single client, it does provide true dynamic memory allocations. The management organizes the available/used memory with three different types of linked lists: the EmptyList contains pointers to the free memory, while the start list holds items with a Bloom filter as well as links to the matching list of records. The list of records holds several addresses and corresponding fingerprints, locating existing items if performed by iterating over successive Bloom filters until something is found and then checking the fingerprint. However, due to the probability-based construction of the data structure using recursive Bloom filters, this manager is not suitable for hard real-time applications.

*Low Latency Hardware-Accelerated Dynamic Memory Manager for Hard Real-Time and Mixed-Criticality Systems* by Kohútka et al. presents a worst-fit approach at dynamic memory management for real-time applications [48]. Available blocks of memory are sorted by size in a max queue. For every allocation request, the largest available block is used, trimming excess memory into a new, smaller block. On deallocation, the deallocated block is recombined with neighboring free blocks (if available). Defragmentation of free blocks separated by allocated blocks is not addressed. An important limitation of the system is the size of the queue, which has to be large enough to hold all free blocks in case of maximum fragmentation. In the worst-case scenario, every second block is allocated at minimum blocksize—resulting in the queue holding entries for half the available addresses.

*Hardware Dynamic Memory Manager for Hard Real-Time Systems* by Kohútka et al. presents the same system again, this time using their *rocket queue* [41] to reduce the large amount of comparators required for the massive sort queue by sharing a smaller set of comparators in a tree-like structure. This comes at the price of more multiplexers and an additional *subtree* counter per queue item [49]. At first sight, this seems to result in *more comparators* since the counter values have to be compared at each level to decide which *subtree* has the least amount of items. However, the examples from [41] show how using counter widths much smaller than the data width can result in overall resource reduction where only a smaller number of queue items are required.

In *Hard Real-Time Memory-Management in a Single Clock Cycle (on FPGAs)* [50], we presented a memory management system specially optimized towards FPGAs, which will be used as the theoretical base for this paper to build upon. It divides the memory space into linkable nodes of equal size called *memory cells*, each having a *pointer* (linking to other cells) and a *data section* containing its payload (which can hold at least one additional pointer called a *data-pointer* for more complicated linking used in our real-time database application). Each memory cell is either under user control or part of the *list of unused cells*, a linked list stored in the unused cells themselves. Apart from that, the only management

data consist of two counters $c_{free}$ (free cells) and $c_{rbnu}$ (reserved but not used cells) and the pointer *ptrUnused* to the *list of unused cells*.

The process of allocating memory is divided into *reservation*—marking that a certain amount of memory has been reserved—and *picking up* the actual memory cells one by one from the *list of unused cells*, passing control over the cell to the user. This differs from the widely used *allocation* approach, where reservation, providing a matching contiguous block of memory (may include time-intensive memory defragmentation) and passing control over to the user are completed in a single operation.

**Remark 2.** *Picking up the reserved memory cells one by one instead is not as much of a practical restriction as it might seem as all memory addresses have to be written individually anyway. This* complete allocation on write *approach matches how memory is actually used: addresses should only be read if they have been written before (apart from special control registers that do not take part in memory management), and writing requires prior allocation. An allocation that is not written to is therefore purely virtual—we do not have to go to the trouble of actually searching and assigning a free memory cell to it. Depending on the user's data structure, this approach can, however, impose additional efforts to remember the addresses of the individual memory cells since address calculation schemes valid for contiguous memory are not applicable.*

Apart from reservations, picking up cells, and returning them, ref. [50] notably provides the special command *ReturnFreeCellRingByAddress(c)*, which returns an arbitrary amount of memory to the manager in $\mathcal{O}(1)$ as long as $c$ contains the cell count in its *data section* and all cells—including $c$—are part of a singly linked ring.

**Remark 3.** *Even though such a ring structure is probably rarely used for all memory structures (as in the real-time database prototype for which we designed the memory manager), we consider it a justifiable extension where hard real-time is required and linked lists are already used.*

### 3. System Model

*3.1. Notation*

Our data structures are built of linked *nodes*. We distinguish two node types:

**Control Nodes** N are used to manage the data structure.

**Data Nodes** N contain user data in their *data* field.

In hierarchical structures, a node can be a control node and a data node at the same time. Denoted as $D_X A_Y$, $D_X$ is the name of the node viewed as data node, while $A_Y$ is the same node's name viewed as control node. Accessing a node's content is denoted as *nodename.fieldname*: A.*next* is the *next* pointer of control node *A* and B.*data* is the *data* field of data node *B*. Pointers hold the address of a node and can either be contained in the *next* field ⟶ or the *data* field ⇢. *NULL*-pointers are marked with ⊢.

*3.2. Memory Organization*

3.2.1. Memory Hardware

In our RTDB prototype, dynamic random access memory (DRAM) and static random access memory (SRAM) have been used as storage. While DRAM is probably the only reasonable choice in practical applications due to speed, size, and cost considerations, any type of memory with bounded response time can be used theoretically.

3.2.2. Memory Management

For memory management, we use the management approach presented in *Hard Real-Time Memory-Management in a Single Clock Cycle (on FPGAs)* by Lohmann and Tutsch [50]. It organizes the memory space into identical entities called *memory cells*, each containing a *next* field (to link several cells together) and a *data* field (for the payload). The manager

keeps a *list of unused cells*, which is stored in the cells it contains, as well as two counters tracking the amounts of currently reserved but not used ($c_{rbnu}$) and free ($c_{free}$) cells.

It offers the operations *RequestReservation* and *ReturnReservationByAmount* for handling reservations, *PickUpFreeCell* (directly requesting a memory cell, may fail if memory is low), *PickUpReservedCell* (identical, but guaranteed to succeed due to previous reservation), *ReturnFreeCellByAddress* (returns a currently used memory cell), and most notably *ReturnFreeCellRingByAddress*, which deallocates arbitrary amounts of memory in $\mathcal{O}(1)$.

While most operations just transfer single memory cells in or out of the *list of unused cells* and update counter values, *ReturnFreeCellRingByAddress* operates on multiple cells at once. This is possible in $\mathcal{O}(1)$ because the memory returned is required to be organized in a particular ring structure shown in Figure 1a: As **L** contains the amount of nodes as well as a pointer to the next node, a single read on it retrieves all information needed without iterating over the cells: the ring is split after **L**, and the resulting list is appended to the *list of unused cells*, which boils down to attaching its first node and leaving the rest untouched, apart from **L**, which is marked as the new end of the *list of unused cells*. The count stored in **L**.data is used to update the counters accordingly.

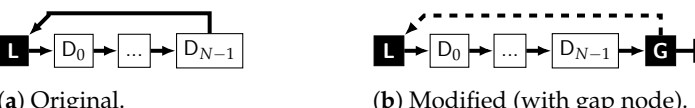

(**a**) Original.  (**b**) Modified (with gap node).

**Figure 1.** Ring passed to *ReturnFreeCellRingByAddress*. L holds the number of nodes in the ring.

**Remark 4.** *Although the user is theoretically able to divert those fields from their intended use as long as the cell is allocated, adopting the memory manager practically implies usage of a linked node structure as the manager does not offer contiguous memory allocations over multiple cells.*

### 3.2.3. Modifications to the Memory Manager

We require the *data* field to be large enough to hold a data pointer, which will be utilized by most of our control nodes. We also introduce the *gap node* **G**, which contains *NULL* in its *next* pointer but always links to another node via its *data* pointer. Its purpose is to serve as a marker in our data structure, similar to a NULL pointer at the end of a singly linked list. We extend the memory manager by allowing gap nodes to be contained in the ring passed to *ReturnFreeCellRingByAddress* (Figure 1b) and in its *list of unused cells* (Figure 2): if the manager processes a gap node, it will follow its *data* pointer instead of *next*. Memory cells originally marked as the last element in the manager's *list of unused cells* with *next* = *NULL* will instead be marked by *next* = *data* = *NULL*. This can always be distinguished from gap nodes as those have *data* $\neq$ *NULL*.

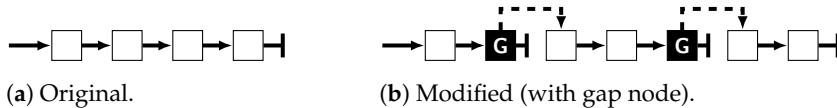

(**a**) Original.  (**b**) Modified (with gap node).

**Figure 2.** The manager's *list of unused cells*.

### 3.3. Hard Real-Time Databases vs. Unknown Amounts of Data

#### 3.3.1. Database Content

- *Insert inserts a new row of data into an existing table.*
- *In a hard RTDB, a transaction's execution must not exceed its deadline.*

As we have to insert a previously unknown amount of row data into our table, an unsolvable problem seems to arise: processing unlimited amounts of data in limited time is impossible. *Luckily, we actually do not have to:* the bandwith between user and RTDB combined with the deadline set by the user imposes fixed limits on the amounts of data that could possibly be transferred. Our processing system therefore qualifies as hard real-time if it can keep up with a continuous stream of data, i.e., the processing of each individual

chunk in the stream is $\mathcal{O}(1)$ in the worst case. So, while processing of *insert* is still $\mathcal{O}(n)$, properly restricting $n$ is up to the user, not the RTDB.

**Remark 5.** *The width of the stream bus (chunk size) is recommended to be an integer multiple of the memory cells* data *field. Refer to Section 7.2.1 on the tradeoff for different* data *field sizes.*

3.3.2. Database Schema

Similar considerations apply to the schema of the database: while the amount of tables, columns, and rows is not restricted by the RTDB itself, the user implicitly introduces limits on those by setting deadlines on related queries or adding new queries, which propagate down to memory limits on certain DB elements, which in turn limit the amount of time required to iterate over them.

**Remark 6.** *For our purposes, parts of the schema currently involved in real-time queries are expected to be static during the query's execution.*

*3.4. Limitations*

The system presented in the following has some limitations: while the higher layers of the hierarchy can be traversed in any direction, movement in the lowest layer is restricted to either hierarchic ascent or forward traversal of the data nodes.

**Remark 7.** *Omitting reverse-traversal in the lowest layer is intentional and cuts down the memory footprint (see Section 7.2.1). As a read operation on a table cell normally reproduces the inserted data in their original order; the authors do not consider reverse reading of the lowest layer (chunks making up the data of a single table cell) particularly useful.*

While we can identify the root element from arbitrarily large structures, there is no analogous way to detect a leaf element without knowledge of the hierarchy depth. If $\mathcal{O}(1)$ is required, deletion of a hierarchy requires (user-defined) limited length of all contained elements above the lowest hierarchy level; i.e., in a table, row and column count are bounded, but data length in the cells is not. As with linked lists (which our structure is based upon), finding the n-th entry in an element or searching for a specific value boils down to a linear search. This paper does not address the problem of performing such search operations in $\mathcal{O}(1)$. We do, however, provide $\mathcal{O}(1)$ primary key access to every known element. For this, we assume that the primary key's value can be generated by the system instead of being chosen by the database user. Our operations do not support composite keys or true set behavior (where all columns are included in the composite primary key). Our solution does not address the topic of storing the data persistently; we consider this a separate problem that is well-covered in the database field.

**4. The Doubly Linked Tree of Singly Linked Rings**

*4.1. The "Element"*

4.1.1. Requirements

The structure of the basic building block—the *element*—is deliberately designed for usage with the hard real-time memory management approach previously mentioned. We choose our *nodes* to be of equal size as one *memory cell*.

For hard real-time deletion, we expect the base elements of our data structure to be connectable in a doubly linked fashion. Otherwise, we would have to run search operations to find the predecessor of the element being deleted as its *next*-pointer must be modified for deletion. Expecting only *delete the element after x* operations (which would be easy on singly linked lists) is not a practical alternative as it passes the search on to the user.

The element's data section is permitted to be constructed of singly linked nodes: we do not consider delete operations on arbitrary data chunks *inside* an element's data section a particularly useful application (unless other elements are contained; see Section 4.1.3).

Regular databases often allow defining arbitrary primary keys, a feature we deem useful for human interaction but not necessary for the database operation itself. Especially in hard real-time environments, where human interaction is pretty much nonexistent (apart from emergency brakes and the like, which are not expected to involve databases), numeric primary keys are dominant. While the probably most used type of primary key is an automatically incremented integer, ascending integer ordering is not required for functionality but chosen as a simple way to generate unique primary keys. The actual requirement is just some kind of unique identifier. For our primary keys, we will take a different approach: by requiring the address of elements to stay constant during the element's lifetime, we can use the element's address as its primary key, entirely skipping the task of key lookup.

### 4.1.2. Storage

To store the actual content of a *doubly linked tree of singly linked rings* element *e*, the following nodes are linked via their *next* pointers in order of appearance (Figure 3):

**A**  The *anchor* provides a stable address, while the element's content might change over time. *The address of the element* will be used synonymous to the anchor's address.

**H**  The *hierarchical node* allows hierarchy ascent and reverse reading. Its data pointer creates a link to the previous anchor and is also called *hierarchical pointer of the element*.

**L**  The *cyclic length node*, containing the number of nodes in the cyclic part of the element, which can be derived either as $L_{cyclic} = N + 3$ (the data nodes plus the three control nodes in the loop) or $L_{cyclic} = length(e) - 1$ (count all nodes, subtract the anchor, which is not part of the loop). It allows usage of the hard real-time memory management from [50], which is also where counting only the cyclic part originates from.

**D**  As many *data nodes N* as required for storage of the element's content in their datafield. This part shall be omitted for an explicitly *empty element* where $N = 0$ (Figure 3b).

**G**  The *gap node*, marking the end of the data section of the element and closing the loop back to the hierarchical node via its data-pointer. Its *next*-pointer is always NULL. Therefore, the module reading an element does not have to count nodes and permanently compare this to $N - 3$ while iterating just to detect the end of the data section.

The nodes of an element *e* are denoted by $e_{\text{L}}$.

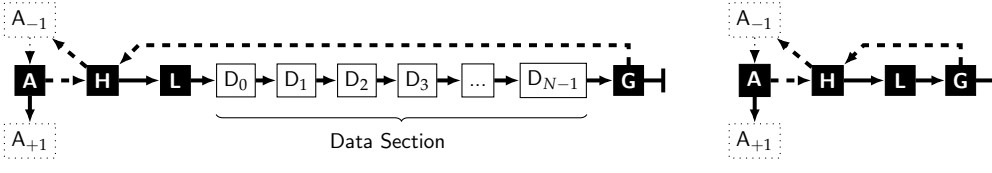

(**a**) Element containing data.                                                            (**b**) Null element.

**Figure 3.** Storage of an element *($A_{-1}$ and $A_{+1}$ hint at possible links from/to other elements)*.

**Remark 8.** *The gap node* **G** *could actually be omitted if the element's data will exclusively be read starting from the anchor, additional hardware usage for counting and comparing is not of concern, and arbitrary entry cyclic access (see Section 4.5.1) is not used. The loop is then closed by the last data node (or the cyclic length node for null elements) pointing to the hierarchic node.*

**Definition 1.** *The **length** of an element e is the number of nodes it is built of. Length is non-recursive: it does not consider the additional nodes of other elements possibly contained in its data section.*

**Definition 2.** *The **content** of an element refers to the pure payload (stored in the $\boxed{D}$ nodes), excluding any metadata (e.g., pointers) that might be used to properly store the element in memory.*

**Definition 3.** *An **empty element** is an element with empty data section ($N = 0$).*

**Definition 4.** *The **null element** $e_\varnothing$ marks a reference explicitly linking to nothing where an element could be. If a pointer points to a null element, its value is NULL.*

**Definition 5.** *The **first data node** $\boxed{D_0}$ in an element is the node in the element's data section, found by following $\boxed{L}$.next. If there are additional nodes in the data section, they are numbered in ascending order starting from the first node to the **last data node** $\boxed{D_{N-1}}$.*

### 4.1.3. Hierarchical Linking

A parent element $e_p$ (denoted $e_c^+$ from the child's perspective) may contain child elements $e_c$ in its data section. In this case, the hierarchic link $e_{c,\boxed{\mathbf{H}}}$ comes into play: it stores a link to the previous element in $e_p$'s data section (wrapping around the ring; itself if it is the only element). This allows reverse reading of the data section of $e_p$. It also introduces the possibility of hierarchic ascent from a child $e_c$ to its parent $e_c^+ = e_p$ via $e_p$'s predecessor.

If *e* has no parent, it is called *root*. Elements without children are called *leaf* elements. In a hierarchy, the root element can be identified by its anchor's next-pointer, referring to the anchor itself. This is a unique property since every other next-pointer will either point to a different node or to NULL. Identifying leaf elements is only possible with a priori knowledge of the hierarchydepth (If required, such hierarchy information (and other metadata) could be included either at the root node or as additional node per element, for example, linked from a control node $\boxed{\mathbf{M}}$ placed directly after $\boxed{\mathbf{L}}$) as we cannot detect if the data nodes in an element contain anchors of an additional hierarchy layer or are simply filled with user data.

**Remark 9.** *For our application—the storage of tabular data in a relational database—the hierarchy depth is fixed by design: #{Database, Table, Row, Cell}.*

### *4.2. Practical Example: Building a Table*

**Remark 10.** *The described element can be used for way more intricate cross-linking: elements can contain links to themselves, multiple elements can point to the same data for deduplication, cross-linking enables mutual communication between otherwise disconnected objects, elements can form two-dimensional grids with their two external pointers ($\boxed{\boldsymbol{A}}$'s next-pointer and $\boxed{\boldsymbol{H}}$'s data-pointer) instead of doubly linked rings. This would also be applicable in our database context, for example, for creating a table index or view as a second version of the higher hierarchy level where the same contained data elements are linked in a different order. However, we want to keep things simple here, and table data is a widely used application.*

### 4.2.1. Hierarchy

The basic hierarchy of a stored table is shown in Figure 4a: all table content is stored in an element $e_{table}$, which contains an element $e_{row}$ for each row. Every row contains a table cell element $e_{cell}$ for each column in the table. Each cell in turn contains the table cell's data in as many $e_{chunk}$ nodes as required for storage. *NULL*-Entries in a table's cell (containing no data) are represented by empty elements.

Exemplary of the data structure, the table is shown as root element. In practical implementations, additional hierarchy layers for the sets of tables and databases are to be expected above, with the list of databases being root.

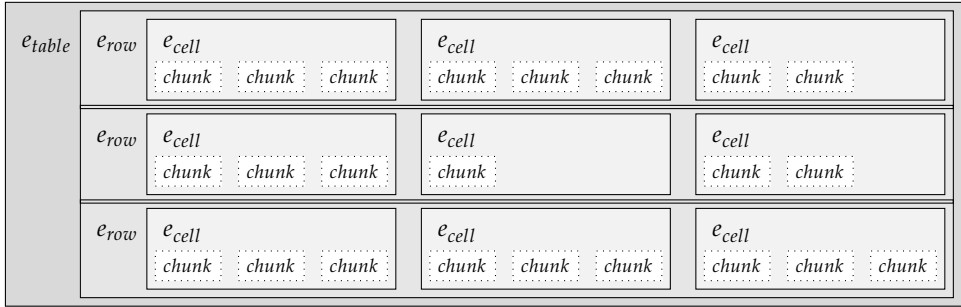

(**a**) Hierarchic relationship of the different Elements *e* and the contained chunks.

(**b**) Available travel directions (logical).

(**c**) Storage at node level.

**Figure 4.** Example: storing a 3 × 3 table with varying chunks per cell (table as root).

4.2.2. Linking Elements

If we now apply the hierarchical approach from Section 4.1.3, we obtain the structure shown in Figure 4b. As we can see, all the anchor nodes are now logically accessible in a doubly linked manner, while the data nodes at the lowest hierarchy level are singly linked.

Figure 4c is the same table drawn at node level. As we can see, the doubly linked characteristic of the top levels is not as direct as shown in Figure 4b for the link to the previous element. It requires traveling from one anchor **A** to its hierarchical node **H** and from there to the previous anchor. This means traveling backward takes twice the time as traveling forward. As this is a constant factor, it does, however, not impact the real-time capabilities of the system.

*4.3. Base Operations*

We distinguish between *base operations*, which stand on their own, and *derived operations* (see Section 4.4), which can be constructed from already discussed operations. This improves readability of the proofs of their respective hard real-time capabilities.

**Definition 6.** *Empty(e) evaluates to* true *if the data section of the element e is empty (N = 0) and to* false *if not.*

As $N$ is not stored directly in the element, this translates to checking if $e_{\boxed{L}}.data = 3$.

**Definition 7.** *Successor(e) returns the next element, which is accessible from $e_{\boxed{A}}$ via its next-pointer. This includes the special case of following $e^+_{\boxed{G}}$ to $e^+_{\boxed{H}}$ to $e^+_{\boxed{L}}$'s next-pointer to wrap around from the last child of $e^+$ to the first.*

**Definition 8.** *Predecessor(e) returns the element, which can be found by following e's hierarchical pointer $e_{\boxed{H}}.data$.*

The predecessor of any element can be found through its associated hierarchic link node **H**. To accomplish this, we follow the associated anchor's data-pointer to **H**. Its data-pointer is the address of the predecessor of our original element. Unlike the *Successor* operation, special wrap around handling at the end of the parent's data section is not required for *Predecessor* as the hierarchical link is always directly connected to an element.

**Remark 11.** *If b is the successor of a, then a is the predecessor of b.*

**Remark 12.** *Predecessor(e) = e = Successor(e) when e is the the only child of its parent $e^+$.*

**Definition 9.** *FirstChild($e_p$) returns the null element $e_\varnothing$ if the parent $e_p$ is empty and the first child element of $e_p$ if it is not.*

This is done by following the parents anchor **A** to its **H** and from there to **L**. If **L**'s data field is zero, we have detected an empty parent and return the null element $e_\varnothing$. If it is not, **L**'s next-pointer is returned.

**Definition 10.** *InsertAfter($e_p$, $e_{ref}$, $e_{new}$) inserts $e_{new}$ into $e_p$ directly after $e_{ref}$.*

Algorithm 1 shows an implementation of Definition 10.

---

**Algorithm 1:** InsertAfter($e_p$, $e_{ref}$, $e_{new}$)

---

$ptrNext \leftarrow e_{ref,\text{[A]}}.next$;
$ptrPrev \leftarrow e_{ref}$;
$e_{new,\text{[A]}}.next \leftarrow ptrNext$;
$e_{new,\text{[H]}}.data \leftarrow ptrPrev$;
$e_{ref,\text{[A]}}.next \leftarrow e_{new}$
$e_{p,\text{[L]}}.data \leftarrow e_{p,\text{[L]}}.data + 1$;

---

*4.4. Derived Operations*

**Definition 11.** *LastChild($e_p$) returns $e_\varnothing$ if $e_p$ is empty and the last child element if not.*

Algorithm 2 shows an implementation of Definition 11.

---

**Algorithm 2:** LastChild($e_p$)

---

**if** *Empty($e_p$)* **then**
│   return $e_\varnothing$;
**else**
│   return *Predecessor(FirstChild($e_p$))*;
**end**

---

**Definition 12.** *InsertAsFirst($e_p$, $e_{new}$) inserts $e_{new}$ into parent $e_p$ at the first position.*

Algorithm 3 shows an implementation of Definition 12.

---

**Algorithm 3:** InsertAsFirst($e_p$, $e_{new}$))

---

$ptrNext \leftarrow e_{p,\text{[L]}}.next$;
**if** *Empty($e_p$)* **then**
│   $ptrPrev \leftarrow e_{new}$;
**else**
│   $ptrPrev \leftarrow LastChild(e_p)$;
**end**
$e_{new,\text{[A]}}.next \leftarrow ptrNext$;
$e_{new,\text{[H]}}.data \leftarrow ptrPrev$;
$e_{p,\text{[L]}}.next \leftarrow e_{new}$;
$e_{p,\text{[L]}}.data \leftarrow e_{p,\text{[L]}}.data + 1$;

---

**Definition 13.** *InsertAsLast($e_p$, $e_{new}$) inserts $e_{new}$ into $e_p$ at the last position.*

Algorithm 4 shows an implementation of Definition 13.

---

**Algorithm 4:** InsertAsLast($e_p$, $e_{new}$)

---

**if** *Empty($e_p$)* **then**
│   *InsertAsFirst($e_p$, $e_{new}$)*;
**else**
│   *InsertAfter($e_p$, LastChild($e_p$), $e_{new}$)*
**end**

---

**Definition 14.** *InsertBefore($e_p$, $e_{ref}$, $e_{new}$) inserts $e_{new}$ into $e_p$ directly before $e_{ref}$.*

Algorithm 5 shows an implementation of Definition 14.

---

**Algorithm 5:** InsertBefore($e_p$, $e_{ref}$, $e_{new}$)

**if** $e_{ref}$ *is FirstChild($e_p$)* **then**
  | *InsertAsFirst($e_p$, $e_{new}$)*;
**else**
  | *InsertAfter($e_p$, Predecessor($e_{ref}$), $e_{new}$)*;
**end**

---

**Definition 15.** *FreeElement(e) frees the memory of e.*

This requires two calls to the memory manager: one call returns the cyclic part of $e$, and the other one returns **[A]** (not contained in the cyclic part). Algorithm 6 shows an implementation of Definition 15.

---

**Algorithm 6:** FreeElement($e$)

ReturnCellByAddress($e$**[A]**);                                    /* returns **[A]** */
ReturnCellRingByAddress($e$**[L]**);                    /* returns the other nodes */

---

**Definition 16.** *DeleteChild($e_p$, $e_c$) removes $e_c$ from $e_p$'s data section and frees the memory previously used to store $e_c$.*

This function definition has a hidden recursion as elements can contain other elements. To comply with hard real-time requirements, we have to limit the number of hierarchy levels $h$ and the length of all elements that contain other elements. Note that the length of the leaf elements (not containing other elements but pure user data) is not limited. Algorithm 7 shows an implementation of Definition 16.

---

**Algorithm 7:** DeleteChild($e_p$, $e_c$)—Without Hierarchy Limit

$ePrev \leftarrow$ Predecessor($e_c$);
$eNext \leftarrow$ Successor($e_c$);
$ePrev$**[A]**$.next \leftarrow eNext$;
$eNext$**[H]**$.data \leftarrow ePrev$;
**foreach** *child $e_{subchild}$ of $e_c$* **do**
  | DeleteChild($e_c$, $e_{subchild}$);
**end**
FreeElement($e_c$);
$e_{p,}$**[L]**$.data \leftarrow e_{p,}$**[L]**$.data - 1$;

---

**Remark 13.** *In our table example, we have utilized three hierarchy levels (for a full database concept, we would add layers for a list of databases and a list of tables): $e_{table}$, $e_{row}$, and $e_{cell}$. Deletion of a row is $\mathcal{O}(\#columns)$ in the general case. If we consider the database schema to be fixed during hard real-time operation, the number of columns is a constant, resulting in the complexity dropping to $\mathcal{O}(1)$. This also applies to all upper layers with one notable exception: varying row counts are very common in practical applications. This means we have to impose an artificial restriction for the row count of every table. Since the actual value does not matter (it just moves the achievable deadline of the individual queries back or forth but does not change the real-time capability of the database system itself), we can leave this choice to the designer of the schema.*

**Remark 14.** *While all upper layers have to be limited for $\mathcal{O}(1)$ deletion, the content in the table cells (the lowest hierarchy layer) may be of any size.*

For practical FPGA implementation, we introduce an overloaded version with an additional parameter $h_c$, which is the number of hierarchy levels of $e_c$:

**Definition 17.** *DeleteChild($e_p$, $e_c$, $h_c$) is an overloaded version of DeleteChild($e_p$,$e_c$), which limits recursion depth according to the amount of hierarchy levels $h_c$.*

The modified execution including $h_c$ is shown in Algorithm 8. A short discussion on how an FPGA implementation of this limited depth recursion might look like is provided in Appendix B.

---
**Algorithm 8:** DeleteChild($e_p$, $e_c$, $h_c$)—Hierarchy Limited

---
**if** $h_c > 1$ **then**
  **foreach** *child $e_{subchild}$ of $e_c$* **do**
    | DeleteChild($e_c$, $e_{subchild}$, $h_c - 1$);
  **end**
**end**
FreeElement($e_c$);
$e_{p,\blacksquare L}.data \leftarrow e_{p,\blacksquare L}.data - 1$;　　　　　　　　/* decrement parent length */

---

**Definition 18.** *Update($e_{target}$, $e_{data}$) deletes the data section of $e_{target}$ and replaces it with a reference to $e_{data}$, preserving the address of $e_{target,\blacksquare A}$.*

As Algorithm 9 shows, this is done by swapping the $\blacksquare A.data$ pointers of the two elements and copying the hierarchic link of $e_{target}$ to $e_{data}$. The hierarchic links do not have to be swapped since the old data will be deleted and *FreeElement()* is independent of the hierarchic link.

---
**Algorithm 9:** Update($e_{target}$, $e_{data}$)

---
$targetPrev \leftarrow e_{target,\blacksquare H}.data$
$originalTargetH \leftarrow e_{target,\blacksquare H}$
$originalDataH \leftarrow e_{data,\blacksquare H}$
$originalDataH.data \leftarrow targetPrev$
$e_{target,\blacksquare A}.data \leftarrow originalDataH$
$e_{data,\blacksquare A}.data \leftarrow originalTargetH$
FreeElement($e_{data}$)

---

**Remark 15.** *The anchor node $e_{data,\blacksquare A}$ of the new element is not strictly required for this operation and could be omitted in practical implementation if an accordingly modified version of FreeElement() is provided. It is still included in the algorithm to preserve this paper's convention of only passing complete elements as parameters.*

*4.5. Other Properties (Not Necessarily Real-Time)*

4.5.1. Arbitrary Entry Cyclic Access

As the content of every element is stored in a cyclic list, it is possible to read the whole content of an element starting at any data node. This might be particularly useful in hierarchic applications as we can perform operations like *read all elements in the same hierarchy level as e* without knowledge of *e*'s parent, essentially providing us with a *get all siblings* operation, which does not depend on hierarchic ascent/descent or having to rewind to the first sibling. Therefore, our database system can support queries like *select all rows that are in the same table as row r* without the need to first look up which table holds *r*.

4.5.2. Hierarchic Ascent

This leads us to the next property: the cyclic nature of the data section with the ring including the hierarchic link allows to ascend in the data structure. This can answer questions like *find the table that contains a specific row* without iterating through all the tables; it is not a top-down search but a bottom-up search; therefore, the non-involved branches of the hierarchy will not be traversed.

## 5. Real-Time Operation (Proof)

*5.1. Prerequisites*

5.1.1. Basic Access

**Assumption 1.** *Read/Write access to a fixed size memory cell/node (given its address) is assumed to be $\mathcal{O}(1)$. This also means following a pointer, e.g., in data or next of a memory cell, has constant access time. We also assume that comparisons between values stored in both fields of a memory cell and increment/decrement operations on these are $\mathcal{O}(1)$.*

5.1.2. Memory Management

**Lemma 1.** *Our modification (see Section 3.2.3) of the real-time memory manager from [50] does not change the memory managers' real-time behavior.*

**Proof.** Our modification allows rings given to *ReturnFreeCellRingByAddress* to contain *gap nodes* ($next = NULL$, with the link to next node instead stored in *data*). This changes the detection of the end of the *list of unused cells* when giving out unused cells to the user: originally, the end was marked by a memory cell with $next = NULL$ — the data field was completely ignored for memory cells in the list of unused cells. With our modification, the end of list condition is now $data = next = NULL$, which can still be checked in $\mathcal{O}(1)$.

Following a gap node in the list of unused cells means that, instead of simply following *next*, we now have to previously check if *next* is *NULL* and in that case follow *data* instead. While this is an additional step, it is still $\mathcal{O}(1)$. □

5.1.3. Storing a Data Stream in an Element

As mentioned in Section 3.3, we assume the arrival of the input as a data stream. The time sacrificed for streaming the data is not introduced by our system but the stream itself, and will therefore not be counted towards the worst-case execution time of our system. To store a chunk of the data stream in the data section of our element, we first need to allocate a limited amount of memory cells/nodes. This is a direct call to the memory managers *PickUpFreeCell* operation, which is $\mathcal{O}(1)$.

**Lemma 2.** *The worst-case time complexity of storing a chunk of finite size to an element is $\mathcal{O}(1)$.*

**Proof.** To store the content of the chunk, we have to allocate ($\mathcal{O}(1)$ according to Lemma 1) and write a finite amount $memoryCellsPerChunk = \lceil sizeof(chunk)/sizeof(memoryCell.data) \rceil$ of data nodes $\boxed{D}$. For the element's overhead, we have to allocate and write additional memory for the cells **A**, **H**, **L** and **G**. This is a finite amount of allocations (each $\mathcal{O}(1)$ when using *PickUpFreeCell()* from [50]) and a finite amount of writes. □

**Corollary 1.** *The worst-case per chunk overhead when storing n chunks in an element, compared to the time it takes to store just the chunks, is $\mathcal{O}(1)$.*

**Proof.** Storing $n$ chunks as raw data is $\mathcal{O}(n)$. With the complexity of storing a single chunk being $\mathcal{O}(1)$ as of Lemma 2, the complexity of storing $n$ chunks in a single element cannot exceed $\mathcal{O}(n)$. This is because, for appending a new chunk to an element, we only have to add the data nodes $\boxed{D_n}$ for the new chunk. Appending a node is, however, just an *insert after x* operation on a singly linked list, which is $\mathcal{O}(1)$. $\Rightarrow$ For every individual chunk, the overhead is limited. This limit is independent of the number of chunks $n$. □

### 5.1.4. Primary Key Access & Reading Data from an Element

**Theorem 1.** *Retrieving the address of an element from its primary key is $\mathcal{O}(1)$.*

**Proof.** The primary key of an element *is* the elements address. □

**Lemma 3.** *The overhead of reading an element's data and sending it as a stream, compared to just sending the data as a stream, is $\mathcal{O}(1)$.*

**Proof.** Iterating through the element involves following the path to the elements **G** node. As there are limited control nodes on this path, the overhead is $\mathcal{O}(1)$. □

### 5.1.5. Accessibility of the Control Nodes

**Lemma 4.** *For an element e, the nodes $e_{\mathbf{A}}$, $e_{\mathbf{H}}$, and $e_{\mathbf{L}}$ are accessible in $\mathcal{O}(1)$.*

**Proof.** The start node of *e* is $e_{\mathbf{A}}$. From there, we can travel along $e_{\mathbf{A}} \dashrightarrow e_{\mathbf{H}} \longrightarrow e_{\mathbf{L}}$. □

### 5.2. Base Operations

**Theorem 2.** *All base operations are $\mathcal{O}(1)$.*

**Proof.** The base operations consist of a limited number of reads and writes to the control nodes **A**, **H**, and **L** as well as incrementing values and following a limited number of pointers. As of Section 5.1.1 and Lemma 4 all of those steps are $\mathcal{O}(1)$. □

### 5.3. Derived Operations

**Theorem 3.** *DeleteChild($e_p$, $e_c$, $h_c$) has a worst-case complexity of $\mathcal{O}(1)$ if the number of hierarchy levels $h_c$ and the length of every non-leaf element is limited.*

**Proof.** One iteration of the implementation Algorithm 7 of *DeleteChild($e_p$, $e_c$, $h_c$)* contains only accesses to nodes accessible in $\mathcal{O}(1)$ (Lemma 4). As the length of the non-leaf-elements is limited, the *foreach* loop has a limited number of iterations. Since the hierarchy levels $h_c$ of *e* are limited, the recursion depth is also limited. □

**Theorem 4.** *All other derived operations except for DeleteChild($e_p$, $e_c$) are $\mathcal{O}(1)$ as well.*

**Proof.** All of them are built of already $\mathcal{O}(1)$-proven operations, node accesses, and following pointers while featuring neither loops nor recursions. □

### 5.4. Storing a Hierarchy

**Theorem 5.** *The worst-case per chunk overhead when storing n chunks in a limited depth hierarchy of elements, compared to the time it takes to store just the chunks, is $\mathcal{O}(1)$.*

**Proof.** This follows from Corollary 1 and *InsertAsLast* being $\mathcal{O}(1)$ (Theorem 4). □

## 6. Experimental Results

### 6.1. Example: The TableWriter

For practical evaluation of our solution, we use the TableWriter, an entity from our hard-real-time database prototype utilizing the doubly linked tree of singly linked rings. The TableWriter receives a data stream annotated with hierarchy information and builds a table from it. The tables anchor address is set via the *configuration stream*. In our prototype, we introduced an additional 'meta node' **M** between **L** and the first data node $\mathbf{D_0}$ (or gap node **G** for empty elements) intended for storing metadata about the individual elements. For our tests, this node was included, but no additional time was spent to write actual metadata to it; as it is contained in the cyclic part of the element, it can simply be viewed as an additional data node.

Figure 5 shows the architecture of the TableWriter:

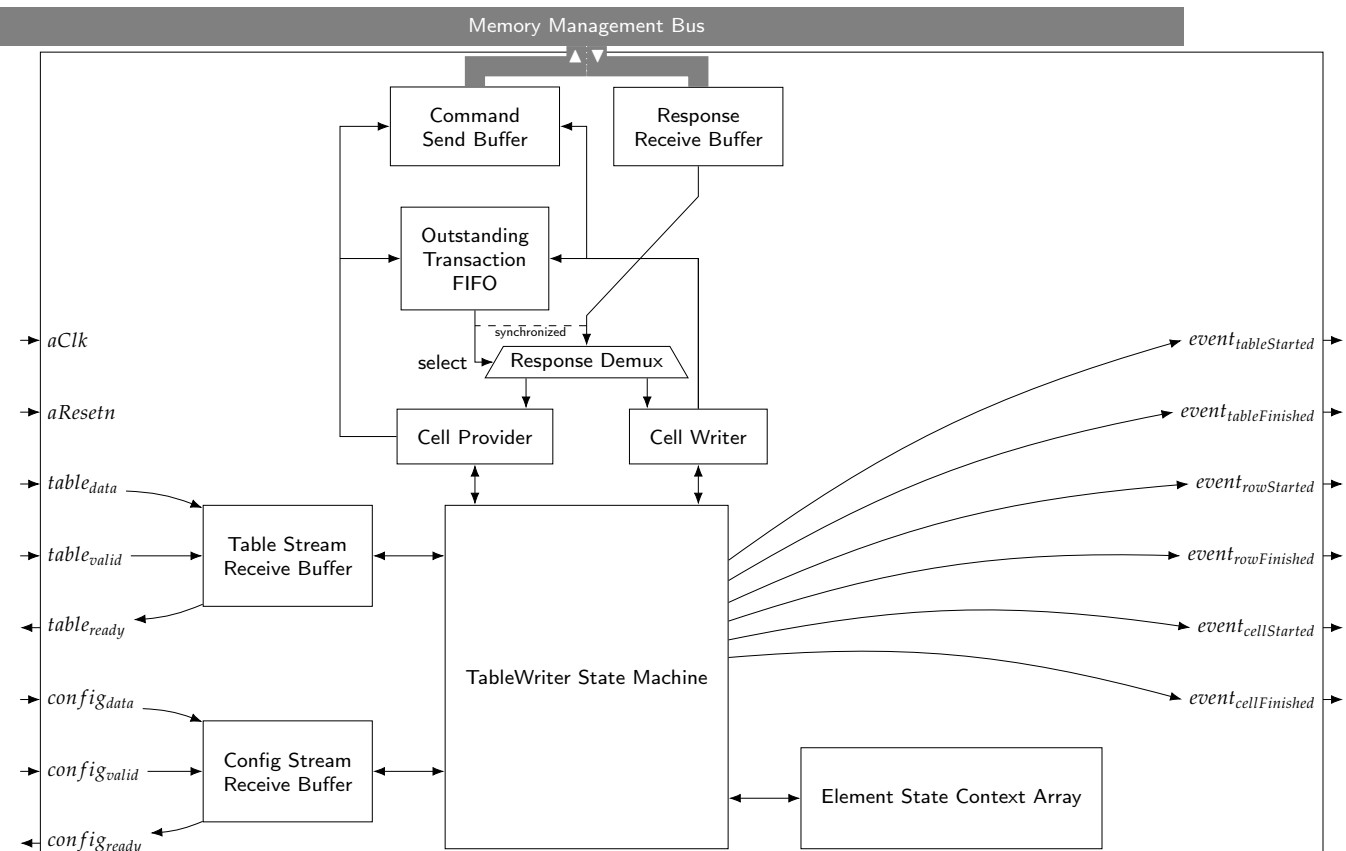

**Figure 5.** TableWriter architecture.

**Table Writer State Machine** The *Table Writer State machine* is the central control unit. It reads the received table data—annotated with hierarchy information in the form *this transfer is the last of the current cell/row/table*—from the *Table Stream Receive Buffer*, which can store a single chunk of data. Its interaction with the central memory management bus (also providing memory read/write) is controlled via the *Cell Provider* and *Cell Writer* units. The state machine itself is implemented as a context switching state machine (as described in Appendix B). The TableWriter uses three contexts—one for each of the hierarchy levels *Table, Row, Cell*. Every context in represents the current operation on an element at this level. It stores the state of the single level state machine (shown in Figure 6), as well as the addresses of the element's anchor nodes, the current cyclic node count, the addresses of a few 'virtual' nodes (*the last node before* **G** *, the first data node, the current data node, the previously processed node, and the first child's* **H** *node*) and finally a boolean determining if the element has been completed.

**Cell Provider & Cell Writer** To simplify interaction with the memory management bus, the TableWriter utilizes two dedicated units to allocate new memory (*Cell Provider*) and write data to memory (*Cell Writer*). The *Memory Provider* simply requests a new memory cell by sending the memory managers *PickUpFreeCell* command and storing the address of the allocated cell returned by the response. When the *Table Writer State Machine* requires a new memory cell, it takes the one previously allocated by the *Cell Provider*, marking the *Cell Provider* as empty—which triggers the *Cell Provider* to request a new memory cell. In contrast, the *Memory Writer* only acts on demand: The *Statemachine* provides a new write task by passing the target address, the data to write and a strobe signal to the *Memory Writer* which decides whether the whole memory cell, only the data field or only the next field shall be written (This was not mentioned

in Section 3.2.3 since it is not necessary for the doubly linked tree of singly linked rings itself. We still use it in the practical evaluation as our prototype already utilizes this updated version of the memory manager (a paper on the updated manager is in preparation). This does not change the systems ability of hard real-time operation since the strobe signal is simply translated to the AXI byte strobe of a single transfer. It does, however, slightly improve the latency, since a partial write would otherwise require knowledge of the non-written content, which can sometimes only be acquired by performing an additional read before writing the cell.) .

**Buffers** The send/receive buffers shown in Figure 5 use a slightly relaxed (We ignore the fact that AXI expects the data field to have a multiple of eight bits as its width.) version of the AXI-Stream protocol by ARM [51]. A buffer $x$ accepts new data if it is empty and the data on $x_{data}$ is marked as valid via $x_{valid} = 1$. Clearing the buffer automatically signals to the outside that new data can be received via $x_{ready} = 1$. This means that transfers happen when $x_{ready} = x_{valid} = 1$.

**Outstanding Transaction FIFO & Response Demux** Since the memory management bus is used both by the *Cell Provider* and the *Cell Writer*, responses have to be directed towards the authors of their corresponding commands. Every time a command is sent to the *Command Send Buffer*, the command's author (CELL_WRITER or CELL_PROVIDER) is inserted into the *Outstanding Transaction Fifo*. When a response arrives on the *Response Receive Buffer*, the *Outstanding Transaction Fifos* output is used to decide where to route the response via the *Response Demux* demultiplexer.

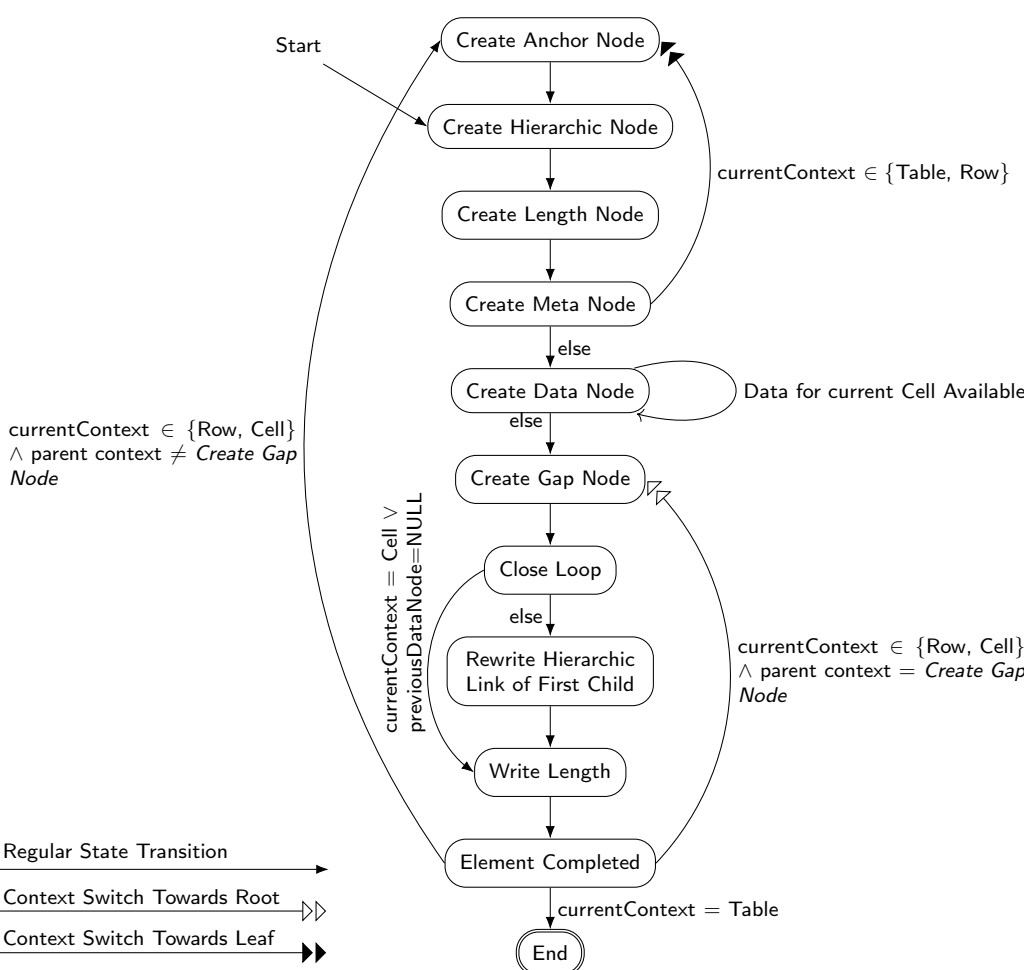

**Figure 6.** Control state machine of the TableWriter.

The actual state machine can be seen in Figure 6. The function of the individual states is described in the following.

**Remark 16.** *For improved performance, the actual implementation pipelines node creation and linking into two steps: Every node is prepared (memory allocation, determining its content) in 'its' state, but written in the following state.*

**Start**  Since the TableWriter receives the anchor node via its *Config Stream* input, the state machine skips the anchor creation state on the table layer and directly starts with creating and connecting the hierarchic node **H** to it in *Create Hierarchic Node*.

**Create Anchor Node**  Creates **A**

**Create Hierarchic Node**  Creates **H** and links **A**.next to **H**.

**Create Length Node & Create Meta Node**  Next, *Create Length Node* creates the length node. At *Create Meta Node* (or *Create Length Node* if you were to implement our original algorithm), the first decision has to be made: if the current context level is not at the leaf layer (*Cell*), we will perform a context switch towards the leaf layer (either *Table* ►► *Row or Row* ►► *Cell*). In the new context, the state machine is initialized (resetting the whole context) to start with a fresh element at *Create Anchor Node*. If instead we are at leaf level, we can start with filling in the streamed data corresponding to the current element by going to *Create Data Node*.

**Create Data Node**  This state creates and appends new data nodes as long as new data are received on the *Table Stream Receive Buffer*. As soon as the received data are marked to be the last chunk in the current hierarchy level (e.g., the last transfer of a *Row*), the state machine proceeds to *Create Gap Node*. If the data are also marked with 'end' markers for higher hierarchy levels (e.g., end of cell + end of row), all affected contexts move to the *Create Gap Node* state as well.

**Create Gap Node**  Here, **G** is created and attached.

**Close Loop**  This step closes the loop/cyclic part of the element. If the current element has at least one child, we have to update the child's hierarchic link in *Rewrite Hierarchic Link of First Child*. Otherwise (either we are a *Cell* level and/or there was not child element), we can directly proceed with *Write Length*.

**Rewrite Hierarchic Link of First Child**  Update the **H** node of the first child. This is only possible after all following childs have been processed since its data-pointer links to the last child's **A**.

**Write Length**  Updates the **L** node to the actual cyclic node count.

**Element Completed**  As soon as an element is completed, the state machine has to decide whether the whole table writing task has been completed (current context is *Table*) or further processing is required. If the context if one of the lower layers *Row or Cell*, we have two possibilities:

> **The parent context expects further data**  In this case (parent context ≠ Create Gap Node), the parent context has not received a corresponding 'your hierarchic level ends here' marker from the *Create Data Node* stage. Therefore, it can directly proceed at the current level by creating the anchor node for the next child of its parent.
>
> **The parent context is about to be finished**  Here, the parent context is already at *Create Gap Node* as the received data marked the current element as the last child of the parent. We therefore continue by switching the context level back up towards the root level (*Cell* ▷▷ *Row or Row* ▷▷ *Table*) and proceeding with the parents gap node creation.

**End**  The whole table has been fully written.

*6.2. Simulation*

6.2.1. Test: Moving Larger Cell in a Square Table

**Condition** In this test, we write equally sized variations of a $4 \times 4$ table to memory. In each run, all cells except a single *focused* cell are of size one, while the focused cell consists of eight memory cells. The testbench performs 16 runs which represent the 16 possible positions of the focused cell.

**Hypothesis** Since all tables consist of an equal count of elements with constant total size, we expect to see identical execution times as well as identical memory usage for all runs. The memory usage is expected to be

$$mem_{table} = (1 + 4 + 16) \cdot \#\{\text{A}, \text{H}, \text{L}, \text{M}, \text{G}\} + 8 + (16 - 1) = 128 \text{ memory cells,}$$

which can be derived as follows: We need one element containing the rows, one for each of the four rows containing the cells and one per cell. The size of each element (ignoring the payload) is $\#\{\text{A}, \text{H}, \text{L}, \text{M}, \text{G}\}$ Since the payload of the leaf elements is always handled equally independent of which element it is contained in, it is possible to simply sum up the individual payloads of the focused $1 \times 8$ and the non-focused $16 - 1 \times 1$ cells. The payload of the non-leaf elements consists exclusively of anchor nodes, which have already been counted in their respective elements.

**Result** Our simulation showed equal execution times of 1333 clock cycles for all runs. The memory consumption after writing the table was always at 128 memory cells.

Figure 7 shows the the current context level of the context switching state machine, the outstanding transactions, and the event outputs of the TableWriter. Although up to 14 outstanding transactions were allowed in the test, the TableWriter only used eight of them at max.

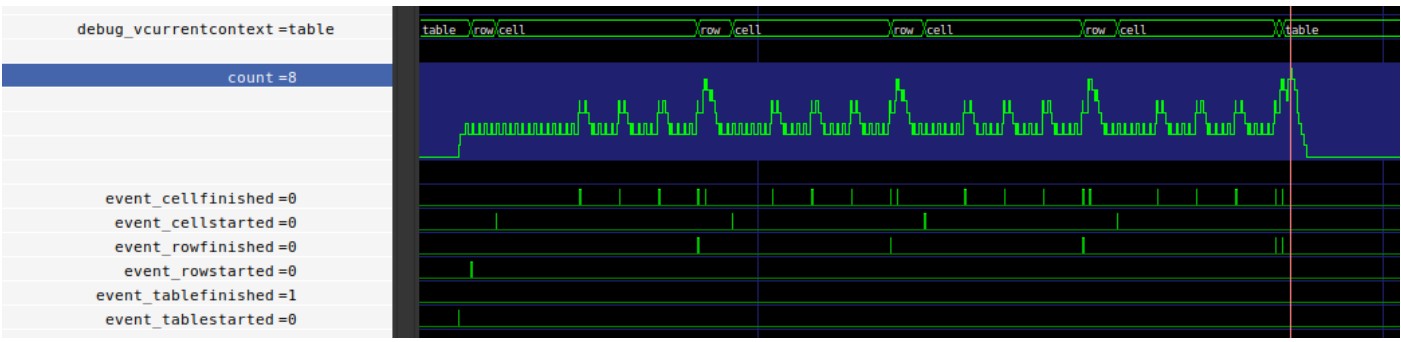

**Figure 7.** Simulation: TableWriter writing a table with four rows and four columns (outstanding transaction FIFO count marked in blue; the values at the left are measured at the red line).

6.2.2. Test: Moving Larger Cell in Non-Square Table

**Condition** Since the first test had an identical number of rows and columns, it might not show problems if the TableWriter would have a preference for square table structures. The following test will evaluate this by applying the same concept to a $7 \times 5$ table.

**Hypothesis** If the TableWriter works correctly it should again show identical executions times and memory usages. Memory usage should be $(1 + 7 + 7 \times 5) \times 5 + 7 \times 5 - 1 + 8 = 257$ memory cells.

**Result** All test runs show identical execution times of 2698 clock cycles. The memory usage has the expected value of 257 memory cells. The maximum outstanding transaction count was again at eight transactions.

6.2.3. Test: Table With Varying Payload and Table Size

**Condition** This time we repeated the first test ($4 \times 4$ table with focused cell), but varied the size of the focused cell. We also completed a second iteration with a $3 \times 4$ table to

test if the table schema has any influence on the execution time required for storing additional payload.

**Hypothesis** Since the execution time for creating the table schema (the higher hierarchy levels) is expected to be independent of the execution time for storing the payload in leaf nodes, we expect linearly rising execution times for the payload with a constant offset for the tables schema. The execution time per additional chunk should be the same for the two different table schemas.

**Result** The simulation results (Figure 8 represents one of the runs) show that the processing time for every additional chunk of payload is exactly 10 clock cycles, independent of the table schema. Memory usage grows exactly by the additional memory cell required to store each new chunk. The $4 \times 4$ table schema's base execution time with one chunk per cell is 1263 clock cycles in our test (Table 1); the $3 \times 4$'s is 766 clock cycles (Table 2).

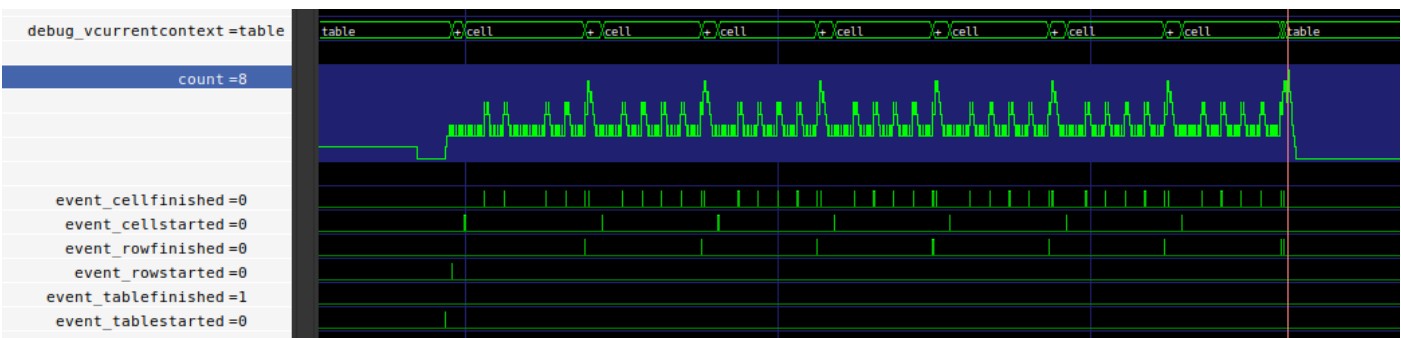

**Figure 8.** Simulation: TableWriter writing a table with seven rows and five columns (outstanding transaction FIFO count marked in blue; the values at the left are measured at the red line).

**Table 1.** Simulation results for a $4 \times 4$ table.

| Focused Cell Size [Memory Cells] | Execution Time [Clock Cycles] | Memory [Memory Cells] |
|:---:|:---:|:---:|
| 1 | 1263 | 121 |
| 2 | 1273 | 122 |
| 3 | 1283 | 123 |
| 4 | 1293 | 124 |
| 5 | 1303 | 125 |
| 6 | 1313 | 126 |
| 7 | 1323 | 127 |
| 8 | 1333 | 128 |

**Table 2.** Simulation results for a $3 \times 4$ table.

| Focused Cell Size [Memory Cells] | Execution Time [Clock Cycles] | Memory [Memory Cells] |
|:---:|:---:|:---:|
| 1 | 766 | 74 |
| 2 | 776 | 75 |
| 3 | 786 | 76 |
| 4 | 796 | 77 |
| 5 | 806 | 78 |
| 6 | 816 | 79 |
| 7 | 826 | 80 |
| 8 | 836 | 81 |

*6.3. Synthesis Results*

*We were asked to include some synthesis results. Please keep in mind those only provide a rough idea of achievable results as developers might have different preferences regarding pipelining and buffer sizes or use other synthesis and implementation tools or settings.*

The test case analyzed accepts a data stream marked with hierarchy information as mentioned in Appendix A. From this, it creates a new table structure resembling the one shown in Figure 4. In this example, the anchor of the table is expected to already exist; its address is received on a second data stream. We use the context switching state machine approach from Appendix B. Our implementation includes some pipelining and a buffer towards the memory manager side to allow for multiple outstanding transactions. The *.next* field of the memory cell has 32 bits. The *.data* field is of identical size, resulting in a memory cell size of 64 bits. The data in Table 3 show synthesis/implementation results achieved on a Xilinx XC7Z045FFG900-2 MPSoC with Xilinx Vivado 2022.2.

**Remark 17.** *The FPGA choice was mainly determined by finding a device close enough to the actual FPGA we use in our project (so the Zynq-Series was set) and the device having enough IO-Blocks (IOBs) so we could actually synthesize the whole TableWriter into it at the top level. Normally, the IOB count should not matter as out-of-context synthesis exists. However, Vivado 2022.2 does not allow out-of-context synthesis for VHDL entities instantiating other entities (heavily used in our design), which is a known limitation. The alternative of synthesizing the TableWriter together with the memory manager, test data creation, database control, etc. (reducing external pin count) was dropped since the synthesis results would be under heavy influence of the connected entities. The required number of LUTs and flip-flops is low enough to fit into any of the FPGAs available in Vivado 2022.2: The smallest devices start at 14400 look up tables (LUT) and 28800 flip-flops (FF). The smallest devices with enough I/O capability (our test utilizes 219 IOBs) start with the XC7Z030 family. We decided to choose the smallest viable device that is also available as part of an evaluation board in the hope of increasing the chance of comparability with other works in the future. This led to the Zynq 7000 ZC706 evaluation board with its XC7Z045FFG900-2 MPSoC.*

**Table 3.** Synthesis results.

| Outstanding Transactions Allowed | Synthesis & Implementation Strategy | Achieved Frequency [MHz] | Worst Negative Slack [ns] | LUT | FF |
|:---:|:---:|:---:|:---:|:---:|:---:|
| 2 | Performance | 166.7 | 0.037 | 3603 | 1487 |
| 2 | Area | 178.6 | 0.000 | 2722 | 1389 |
| 7 | Performance | 161.3 | 0.012 | 3783 | 1501 |
| 7 | Area | 169.5 | 0.005 | 2738 | 1407 |
| 15 | Performance | 153.8 | 0.044 | 3720 | 1520 |
| 15 | Area | 152.7 | 0.025 | 2801 | 1426 |

We completed multiple synthesis runs with increasing clock frequency until we reached negative values for the worst negative slack. Surprisingly, with smaller buffers for outstanding transactions, the vivado run strategies optimized for lower area usage achieved higher frequencies than the run strategy recommended by vivado for higher performance. The large difference in LUT usage between performance and area strategies and the declining frequency with increasing outstanding transaction buffers suggest that our implementation still has potential for optimizations. The unit using the largest amount of FFs is the *Element State Context Array*. FF utilization could likely be reduced as we currently keep track of the addresses of all important nodes in an element (for development purposes), while only part of this information is strictly required to build a new table. Most of the steps in the *Control State Machine* only require information about neighboring nodes, so FF usage could be further reduced by reusing the same FFs for different node addresses.

## 7. Comparison

### 7.1. Our Approach vs. Common Data Structures

As we require hard real-time, *hash tables* ($\mathcal{O}(1)$ access on *average*) are not applicable to our use-case. *Arrays* can be used as lists with $\mathcal{O}(1)$ deletion (by somehow marking empty entries) but do not provide $\mathcal{O}(1)$ insertion at arbitrary positions. Insertion at any but the last position will also inevitably change the address of all following entries. The remaining data structures are, in contrast, able to keep the address of an entry constant over its lifetime as they all follow a node-based approach.

The next structure we can rule out is the *random access list*: while [52] showed that insertion and deletion can be done in hard real time at the front and back of a list, it does not include the necessary operation of deleting an arbitrary entry in $\mathcal{O}(1)$.

*Singly linked lists* provide $\mathcal{O}(1)$ insertion and deletion of a node after a specific entry. Deletion of an entry itself is, however, $\mathcal{O}(n)$ as a linear search has to be performed to find its predecessor (which will be re-linked to the successor of the entry removed). The same goes for insertion before a specific entry. *Doubly linked lists* improve in that regard as the additional pointer to the previous node enables $\mathcal{O}(1)$ access to a node's predecessor. Extending the linked list to a *ring* allows deletion of a whole list at once with the hard real-time memory manager discussed before.

*Trees* are a perfect fit to represent hierarchy. However, an implementation with all nodes being of the same structure (one of our targets) implies setting the leaf node's child pointers to NULL, wasting memory. On the other hand, such an implementation allows deciding if a node is a child by just looking at the node. Supporting operations like *Successor* and *Predecessor* would either require storage of additional pointers to parents (like our approach) and siblings in every node or searching from the root node. Also, the standard algorithms only provide $\mathcal{O}\log(n)$ access times. While it would be possible to link individual nodes to a specific position in $\mathcal{O}(1)$ if there is space for another child, the limited number of $k$ direct children also restricts not only the number of rows and cells but also the chunks of a table. Furthermore, the same limit of direct child nodes would apply at all individual hierarchy levels, which does not represent the practical use case. A typical table in a database will probably have millions of rows but significantly fewer columns.

*Our approach* combines the advantages of singly and doubly linked lists/rings with the hierarchic approach of trees: while the base elements can be connected in a doubly linked fashion for $\mathcal{O}(1)$ deletion, the leaf elements containing the payload are only singly linked, saving memory. Primary key lookup is completely skipped. At the same time, it manages to keep all hierarchy levels the same, which simplifies the algorithms, leading to less hardware usage and higher performance in hardware implementations on FPGAs.

Closing the ring after the data section allows to use the memory manager from [50], which is especially important as it enables deallocation of arbitrarily sized leaf elements in $\mathcal{O}(1)$. This means that, for a table with a limited amount of rows and columns but unlimited data chunks contained in the cells, a complete table can still be deleted in $\mathcal{O}(1)$.

As we split all data into identically sized linked nodes, our system's memory footprint and read/write speed can, however, not compete with systems allowing dynamic data-matched node sizes. It is a solution for hard real-time, not for maximum efficiency.

### 7.2. Memory Consumption

#### 7.2.1. Singly vs. Doubly Linked Rings

Are the efforts for enforcing single linking actually worth it? How does the memory footprint of our approach of single linking the rings compare to a more obvious version where every node is doubly linked? After all, this would additionally allow reverse travel also in the leaf layer, remove the need for **H** (slightly reducing execution times), and provide a simple way to insert/delete individual $\boxed{D}$ nodes in the leaf elements (an operation we did not require for our application, but probably useful in other scenarios.).

The memory consumption of a single element storing *payload* bytes in memory cells with pointer size $p$ and data field size $d$ (in bytes) is calculated by the following formulas:

$$memSLR(payload, p, d) = (p + d) \cdot \left( \left\lceil \frac{payload}{d} \right\rceil + \#\{\text{A}, \text{H}, \text{L}, \text{G}\} \right) \quad (1)$$

$$memDLR(payload, p, d) = (2p + d) \cdot \left( \left\lceil \frac{payload}{d} \right\rceil + \#\{\text{A}, \text{L}, \text{G}\} \right) \quad (2)$$

Figure 9a compares the memory footprint of our approach (SLR) to such an alternate version with doubly linked nodes (DLR), analyzing a single element storing increasing payloads across memory cells with various *data field* sizes and a fixed pointer size of 8 bytes. Since the memory cells *data* field has to be at least as large as the pointer, we start with a data field size equal to the pointer and work our way up in powers of two.

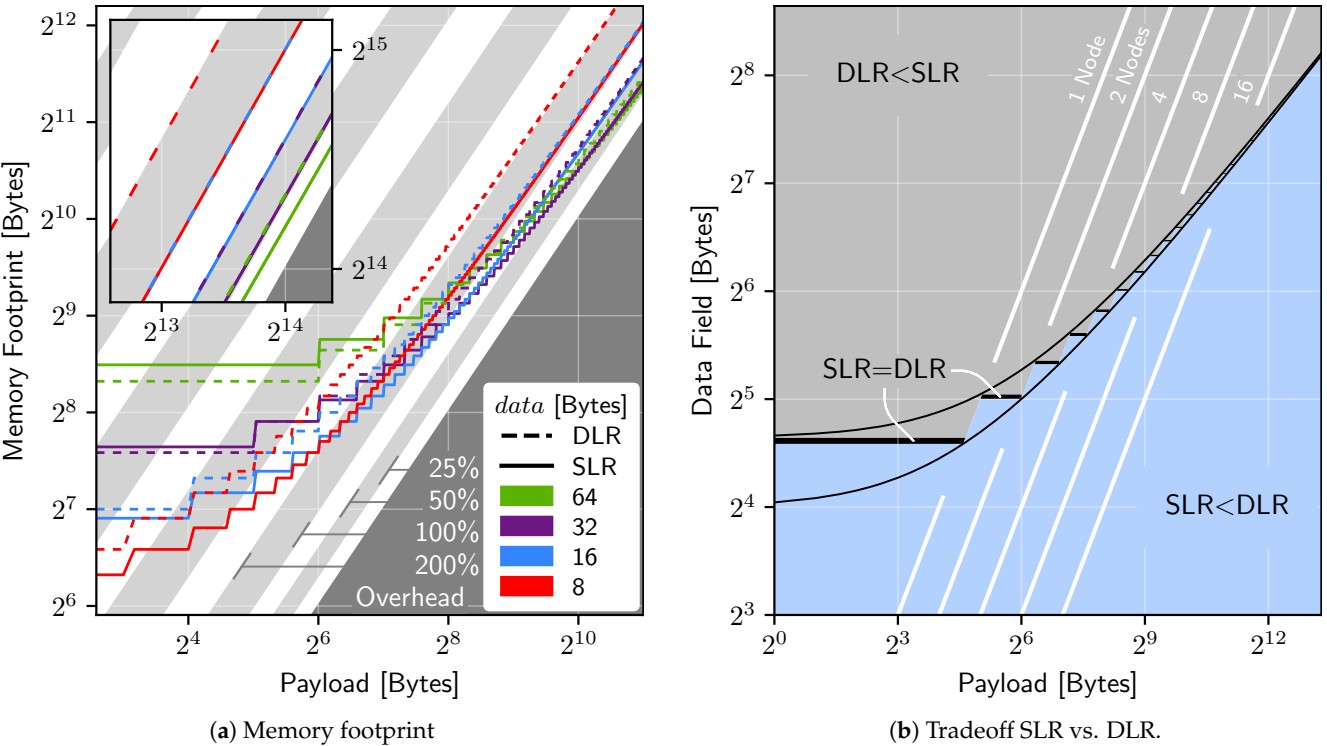

(**a**) Memory footprint                    (**b**) Tradeoff SLR vs. DLR.

**Figure 9.** Single element comparison: singly vs. doubly linked nodes ($p = 8$).

With a small memory cell data field, the SLR 8 model uses significantly less memory than its DLR counterpart (over all payloads). At larger data field sizes, the additional pointer in the DLR is less significant to the total memory consumption than the increasingly underutilized data fields of the control nodes, resulting in less memory usage for relatively small payloads (SLR 32, SLR 64). With increasing payloads, the SLR takes the leading position again as the contribution of the control nodes to the total memory consumption diminishes compared to the payload itself. For increasing payloads, the overhead approaches that of a single memory cell ($p/d$ vs. $2p/d$). The SLR is strictly better for

$$(p + d) \cdot \left( \left\lceil \frac{payload}{d} \right\rceil + 4 \right) < (2p + d) \cdot \left( \left\lceil \frac{payload}{d} \right\rceil + 3 \right) \quad \Leftrightarrow \quad \frac{d}{p} - 2 < \left\lceil \frac{payload}{d} \right\rceil \quad (3)$$

The corresponding areas are shown in Figure 9b. The stepped borders are confined to the area between the quadratic equations $d_l \leqslant d < d_r$ with

$$d_{l,r} = ap + \sqrt{(ap)^2 + p \cdot payload} + b, \quad a = \begin{cases} 1/2 & l \\ 3/2 & r \end{cases}, \quad b = \begin{cases} 1 & SLR \leqslant DLR \\ 0 & SLR < DLR \end{cases} \quad (4)$$

### 7.2.2. Storing a Table

To store a table with $n$ rows, $m$ columns, and a total payload of $x$ memory cells, the doubly linked tree of singly linked rings requires

$$memDLToSLR(n, m, x) = x + (1 + n + nm) \cdot \#\{\boxed{A}, \boxed{H}, \boxed{L}, \boxed{G}\} \text{ memory cells}$$

memory cells. Using singly linked lists or rings instead of our elements would reduce the memory consumption to

$$memLinkedList = x + 1 + n + nm \text{ memory cells}$$

since the control nodes $\boxed{A}$, $\boxed{H}$, $\boxed{L}$, $\boxed{G}$ are being replaced by a single node, which essentially represents the same thing as $\boxed{A}$—a data node in the parent list that stores a link to a lower-hierarchy-level list. The node size for a singly linked list is the same as in our approach. This reduction in memory does of course come at the price of losing some capabilities, e.g., the known node count from $\boxed{L}$ or the detection of the end/start point provided by $\boxed{G}$ in case a ring structure is used.

For a tree, the node size is the sum of the payload and the number of pointers required, e.g., A $k$-ary tree has a node size of $k \cdot p + d$. If the hierarchy in the tree is supposed to represent the actual hierarchy of the table (each parent links to all its childs), the node size has to accommodate for the largest possible tree node. For our table, we obtain

$$k = \max(n, m) \tag{5}$$
$$nodesize_{tree} = \max(n, m) \cdot p + d \tag{6}$$

As already mentioned in Section 7.1, database tables normally have a fairly limited number of columns but may have millions of rows. That already indicates that this approach will have a massive overhead: The many nodes storing the links to the relatively few individual column cells in a row have the same size as the relatively few but large nodes storing the links to all the rows. For the actual payload, one would likely utilize a dynamically sized tree or list structure since holding the whole payload in a single node would worsen this problem even more where large variations in data sizes occur.

The memory required to store the structure of our table is

$$memTreeStructure(n, m) = (1 + n + nm) \cdot nodesize_{tree} \tag{7}$$
$$= (1 + n + nm) \cdot (\max(n, m) \cdot p + d) \tag{8}$$

For the storage of the payload in the optimal scenario where the data stored in the data field of the nodes spreads perfectly over the existing nodes, we can assume

$$memTreePayload(k) = \left\lceil \frac{k}{nodesize_{tree}} \right\rceil \cdot nodesize_{tree} \tag{9}$$
$$= \left\lceil \frac{k}{\max(n, m) \cdot p + d} \right\rceil \cdot (\max(n, m) \cdot p + d) \tag{10}$$

For practical applications, this value is expected to be significantly larger, especially for table cells with a few bytes of data, where the overhead of $k \cdot p$ pointers leads to most of the node storage being overhead.

The total amount of memory required for the tree approach but keeping the original hierarchy is (in the optimal case of evenly spread data)

$$memTree = memTreeStructure + memTreePayload \tag{11}$$
$$= \left( (1 + n + nm) + \left\lceil \frac{k}{\max(n, m) \cdot p + d} \right\rceil \right) \cdot (\max(n, m) \cdot p + d) \tag{12}$$



While the fraction approximately cancels out with the node size to *k* for the payload (keep in mind this is only for the optimal case), *memTreeStructure* introduces the multiplicative term $(\max(n,m))^2$, which means that the node size is growing quadratically with the larger one of the table dimensions, additionally to the term $\max(n,m) \cdot \min(n,m)$, which was expected from the fact that this is a $n \times m$ table.

## 8. Conclusions

Of the discussed list-like data structures, our approach features the largest feature set while keeping almost every operation hard real-time. The only exceptions are n-th element access and recursively deleting a whole hierarchy of elements. The latter is, however, possible in $\mathcal{O}(1)$ if hierarchy depth and the length of every but the lowest layer is restricted. For our target application—a hard real-time database—this means deleting a table with a limited amount of rows and columns, but unlimited data is possible in $\mathcal{O}(1)$—including deallocation time.

More generally speaking, our approach provides a way to build hierarchical data structures with doubly linked behavior in every but the lowest layer, which we consider a good compromise between freely navigating in the structure and keeping a low memory overhead. At the same time, it still keeps the structure of every building block (=element) the same, which simplifies implementation in hardware designs on FPGAs as sequential decisions for each step are minimized. As a result of this, most of our algorithms can be parallelized to a high degree.

This comes at the cost of the whole memory being organized in identical memory cells/nodes, increasing the memory overhead compared to contiguous storage. Insert and update operations can keep up with arbitrarily large incoming data streams, limiting the overhead to $\mathcal{O}(1)$. Using the anchor node's address as primary key of the element allows direct data access without key lookup. As a side note, our approach also provides non-real-time operations like hierarchic ascent and cyclic access, both available from any starting node.

*Further Ideas*

For larger memory cell *datafields*, the information of most control nodes could be directly stored in **A**—and/or multiple anchors could be packed into a single one—reducing the overhead of the elements further. While a hardware design adopting this based on generics/parameters could certainly be created, we expect it to be quite a challenge. Specialized HWDS units could potentially accelerate access times, e.g., storing just control nodes of a specific section for fast rearrangement of elements, or to transparently ensure atomic execution of the operations, which is required for concurrent access. Some preparations for this have been undertaken during the design, e.g., linking elements via **A** and **H**, which allows replacing arbitrary elements with only two write operations. The complete solution for concurrent operation is, however, still a work in progress.

**Author Contributions:** Conceptualization, S.L. and D.T.; Data curation, S.L.; Formal analysis, S.L.; Funding acquisition, D.T.; Investigation, S.L.; Methodology, S.L.; Project administration, D.T.; Resources, D.T.; Software, S.L.; Supervision, D.T.; Validation, S.L.; Visualization, S.L.; Writing—original draft, S.L.; Writing—review and editing, S.L. and D.T. All authors have read and agreed to the published version of the manuscript.

**Funding:** This research received no external funding.

**Data Availability Statement:** The data presented in this study are available in this article (see Sections 6.2 and 6.3).

**Conflicts of Interest:** The authors declare no conflicts of interest.

## Abbreviations

The following abbreviations are used in this manuscript:

| | |
|---|---|
| BLOB | Binary Large Object |
| CPU | Central Processing Unit |
| CRUD | Create, Read, Update, Delete |
| DB | Database |
| DLR | Doubly-Linked Ring |
| DRAM | Dynamic Random Access Memory |
| FF | Flip-Flop |
| FPGA | Field Programmable Gate Array |
| FSM | Finite State Machine |
| HWDS | Hardware Data Structure |
| IOB | Input Output Block |
| LUT | Look-up table |
| RTDB | Real-Time Database |
| SLR | Singly-Linked Ring |
| SRAM | Static Random Access Memory |
| WCET | Worst-Case Execution Time |

## Appendix A. Example Interface

The presented solution is intended to be customized according to the requirements of the developer (e.g., different hierarchy depth, more complex node-linking; see Remark 10) by building on the set of operators presented. For example, in our hard real-time database prototype, we use a set of data processing units utilizing different subsets of the presented operations: some modules only read data, some write or update, while others may only change links in the existing hierarchical structure. The interface shown in Figure A1 may be used as a starting point for customizations.

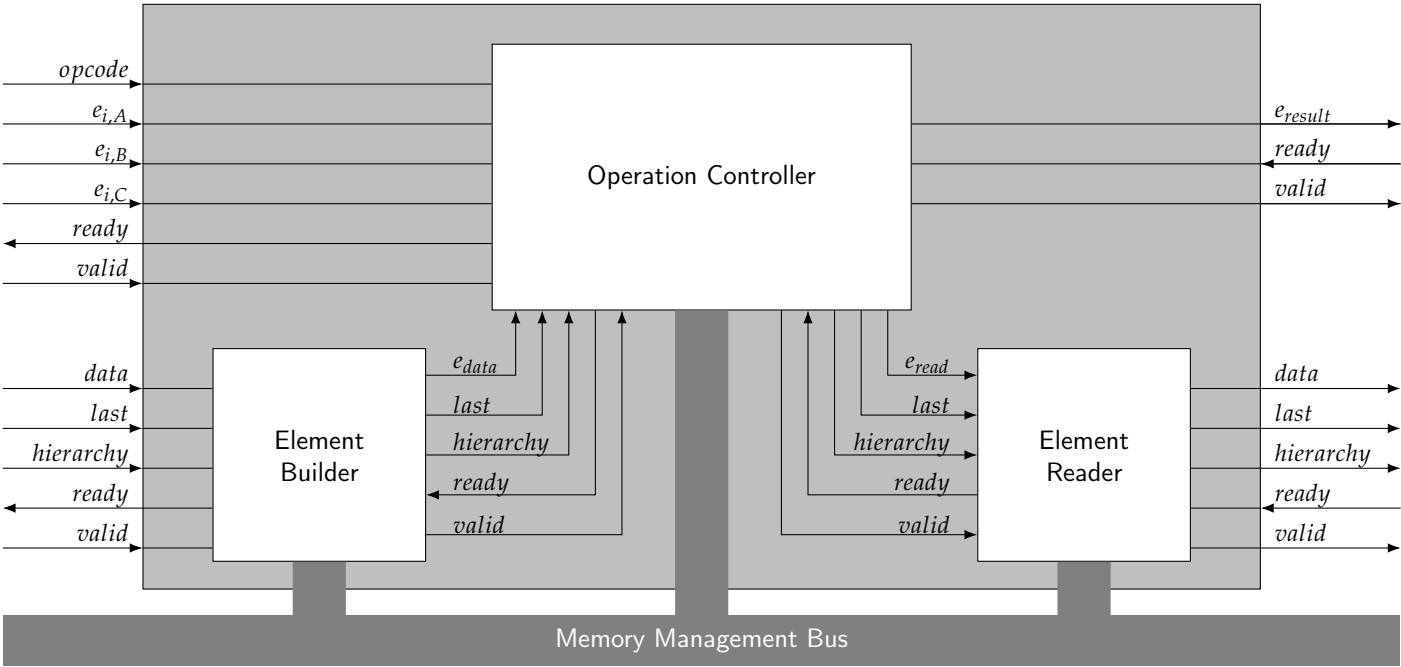

**Figure A1.** Interface logic.

We split the functionality into three submodules: The *Element Builder* accepts a data stream intended for *Insert/Update* operations and stores it to a new element. The *Element Reader* reads the data of an element and sends it as data stream. The *Operation Controller* exclusively operates on already stored elements. It performs all the interlinking between elements. This includes unlinking elements before the *element deallocator* frees their memory.

The operation to be executed is passed to the *Operation Controller* via the *opcode* input. The parameters to the operation (always of type element) are provided on the corresponding $e_i$ ports, with $e_{i,A}$ corresponding to the first, $e_{i,B}$ to the second, and $e_{i,B}$ to the third parameter. If an operation uses less than three parameters, the values of the unused $e_i$ are ignored. The *Operation Controller* returns its results (e.g., the predecessor of a node for the *Predecessor($e_{i,A}$)* operation) on $e_{result}$.

For the *Insert/Update* operations, the user may choose between either linking in an existing element (passed via $e_i$) or creating a new element from a data stream, which is then used instead. For this, the user has to (in any order)

- Send $e_\varnothing$ for the $e_i$ to be replaced
- Send the data to the *Element Builder*

The *Element Builder* will now create an element from the data stream and send it to the operation controller, which will use it for the 'missing' parameter. For example, to execute *Update($e_{target}$, $e_{data}$)* but with $e_{data}$ replaced by the data stream, the user has to

- Send the opcode for *Update*
- Send the data stream to the *Element Builder* via its *data*, *last* and *hierarchy* inputs
- Set the input $e_{i,B}$ corresponding to $e_{data}$ to $e_\varnothing$

The *Element Builder* will now build $e_{data}$ from the stream, which is then used in the *Update* operation. The *Element Reader* is used whenever a data stream has to be output by an operation. In this case, the *Operation Controller* simply sends the element to be read to the *Element Reader*, which will then output the element's content as a data stream on its outputs *data*, *last*, and *hierarchy*.

For I/O transfer synchronization, we adapt the valid/ready handshake of the AXI-Stream protocol [51]. On the *Operation Controller*, the signals *opcode*, $e_A$, $e_B$, and $e_C$ are combined into a single bus. Since each operation takes a limited number of parameters and returns a single element; using valid/ready is sufficient as only single-transfer transactions are required. For the input/output data streams, however, an additional marker to indicate where a cell, row, or table ends is required. This is handled by the *last* (something ends here) and *hierarchy* (is it a table, row, or cell?) signals. Since the hierarchy information is also required between *Operation Controller* and *Element Builder/Reader* to properly handle hierarchies, those signals are copied from the data stream to the element stream and vice versa. All submodules are connected to the memory manager via a central bus handling both memory management and read/write operations (a paper on the manager's architecture and the corresponding memory bus is currently in preparation).

To execute, e.g., a typical relational databases INSERT operation, the user would send the matching command to the *Operation Controller* and simultaneously send the data stream to the *Element Builder*. The *Operation Controller* prepares a new row element $e_{row}$, while the *Element Builder* creates the corresponding child elements from the individual cell data stream. Cell and row ends are marked in the data stream via *tLast* and *hierarchyLevel*. As soon as all content is fully available, the *Operation Controller* will link $e_{row}$ into the target table (if concurrent access is not of concern, this may also be completed directly).

**Appendix B. Synthesizable Implementation of Limited Depth Recursion**

While most of our operations are fairly straightforward to implement in an FPGA for the experienced FPGA developer, one is less obvious: iterating the data structure, e.g., as in Algorithm 8, involves (limited depth) recursion. How can this be implemented in synthesizable logic? We suggest to use something we would call a *context switching state machine*, a finite state machine performing its tasks based on a single context but able to switch to different contexts if it hits certain states. States in the current context may modify the state of the other contexts to prepare for upcoming context switches.

**Remark A1.** *While this could be represented as a regular FSM, the context-switching description of it is much more concise and more closely resembles its recursive nature. It is also closer to how the suggested FPGA design actually works.*

Our implementation features one context per hierarchy level, each storing information on a single element like the addresses of all the already visited control nodes, positional information like the current and the previous node or the current length of the ring, and the task to perform on that level. Transitioning between elements is done by performing a context switch, or, in hardware terms, multiplexing the FSMs state and the corresponding context information. Since element transitions always happen via the **A** and **H** nodes, context switching is limited to neighboring hierarchy levels in our application. This also implies that write access to the other contexts is fairly limited, keeping the logic complexity relatively low. Since the doubly linked tree of singly linked rings treats all hierarchy layers the same, the single-context state machine is identical for every mid-layer context. Some special case handling is required for the contexts representing the root/leaf layer as those are not able to switch to higher/lower layers.

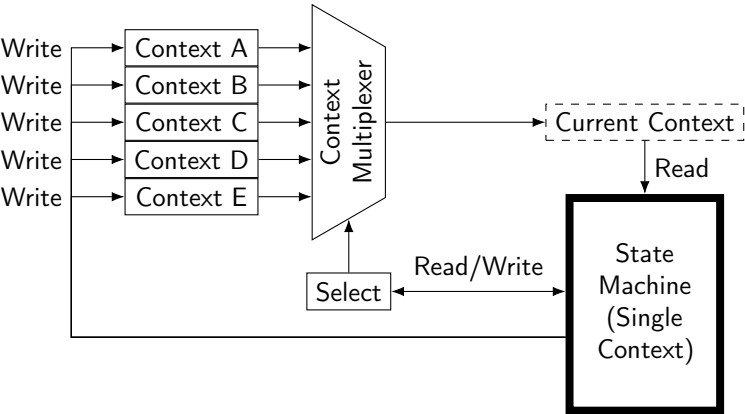

**Figure A2.** Context switching state machine.

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
