# Peer review of "The Doubly Linked Tree of Singly Linked Rings: Providing Hard Real-Time Database Operations on an FPGA"

_computers, doi:10.3390/computers13010008_

Round 1

Reviewer 1 Report

Comments and Suggestions for Authors

This paper is very hard to understand due to its structure. In particular, the introduction should start by explaining the general scenario and the problem that should be solved. In this paper the first paragraph is about the contribution. Moreover, the continuous Remarks and Definitions do not help the reader. This should be a scientific paper, but it is written as a textbook of mathematics! The experimental evaluation of the proposed method lacks of a real life scenario, since it has been mainly performed as proofs of theorems.

The most interesting part of the paper has been related to Appendix C, where the authors briefly shows the synthesis results (without explaining how they implemented the system and how they chose the FPGA devices).

Due to this critical issues, I am not recommending this paper for publication.

Comments on the Quality of English Language

Please, do not use the capital letter after the colon. 

Please, explain the acronyms the first time you use them.

Reviewer 2 Report

Comments and Suggestions for Authors

This paper is very interesting with very relevant topic of research. The paper presents a doubly linked tree of singly linked rings, a hierarchical node based data structure providing the essential database table operations Read, Insert, Update and Delete for hard real-time databases while keeping all elements in all hierarchy levels of identical structure, reducing the amount of sequential decisions (and completely omitting primary key lookup) and thus improving achievable performance on FPGAs.

In addition to the pseudo-code and high-level graphs, the proposed solution should be also described using block diagrams, showing the top-level module and internal submodules, including interfaces (ports of these modules, especially for the top-level module).

The paper is missing verification information (i.e. how the solution was verified), ideally with simulation results and/or waveforms.

The paper is also missing synthesis information, such as which FPGA device was used for synthesis, what maximum clock frequency was achieved without causing negative slack times in Static Timing Analysis, and resource utilization of FPGA (number of consumed LUTs or other resource, ideally with respect to some parameters that can be changed to show scalability of your solution too).

Proposed solution should be compared to existing (HW or SW) solution to demonstrate the advantages and disadvantages of your solution.

Last but not least, research papers in high-quality journals like this are expected to contain more references and better/bigger Related Work part of the paper listing a number of existing solutions. You should add more existing solutions related to this research topic, including memory management in HW, data sorting in priority queues and task schedulers, such as for example:

- SRAM-based architecture of priortiy queues called MultiQueue

- ASIC/FPGA architecture called Heap Queue

- data sorting architecture for real-time systems called Rocket Queue

- Hardware Dynamic Memory Manager for Hard Real-Time Systems

- Low Latency Hardware-Accelerated Dynamic Memory Manager for Hard Real-Time and Mixed-Criticality Systems

- examples of task schedulers:

- A New FPGA-Based Task Scheduler for Real-Time Systems

- A New FPGA - based Architecture of Task Scheduler with Support of Periodic Real-Time Tasks

- Novel efficient on-chip task scheduler for multi-core hard real-time systems

Reviewer 3 Report

Comments and Suggestions for Authors

Please use the common acronym CRUD for your operations. What you are describing is rather a key-value store than a database (record-oriented, not set-oriented). You do not mention the kind of memory you assume for your data structures. Obviously, is is directly and byte adressable, so no harddisk or SSD - persistence?  An important assumption seems to be that access time is always the same, irresp. of the address.  No concurrency yet.

Minor issues:

Line 85: double "to"

Line 198: Shouldn't it be "ReturnFreeCellRingByAddress" here?

Line 466: an element's data

Round 2

Reviewer 1 Report

Comments and Suggestions for Authors

Authors partially addressed my comments.

However, the initial limitations of the paper have not been successfully addressed. My comment “Moreover, the continuous Remarks and Definitions do not help the reader. This should be a scientific paper, but it is written as a textbook of mathematics!” has been addressed as “The paper is (at its heart) a mathematical one ;-)” (with a very unprofessional smile at the end of the sentence). Since this journal is titled “Computers” and not “Mathematics”, the writing style should be the one of the other papers published in the journal. This reviewer suggests to carefully read the latest paper published by Computers to realize that they are not a list of theorems and proofs.

The section “Practical Evaluation” should be renamed as “Experimental results”.

The sentence “The FPGA choice was mainly determined by finding a device close enough to the actual FPGA we use in our project (so the Zynq-Series was set) and the device having enough pins so we could actually synthesize the whole TableWriter into it at the top level” should be extended. Is the number of pins the main feature in choosing the device? What are the main features of the TableWriter and how do they impact on resource consumption?

Reviewer 2 Report

Comments and Suggestions for Authors

In the revised paper, only 1 comment out of 5 was addressed, and even that one was not done properly (missing explanation of individual ports/signals).

The remaining 4 comments were entirely rejected by the authors with inadequate reasoning.

Round 3

Reviewer 2 Report

Comments and Suggestions for Authors

The revised version of paper was improved in multiple points based on my first review. I appreciate that. However, I still see 2 issues in this paper.

Issue #1, the Related Work could be still improved in sections 2.3 and 2.4.

Section 2.3 is referring to HWDS data structures proposed by Bloom, which are mainly used in (priority) queues and schedulers, including data sorting, which is a basic feature used in databases (see SQL queries SELECT). For the schedulers, you have mentioned only one such solution but there are many more papers about schedulers. I still see a big gap in this part of paper (for example MultiQueue presented as "Efficiency of Priority Queue Architectures in FPGA").

Section 2.4 is even more important for this paper, so it is good that this section contains more details. But the very first sentence states "Dynamic memory management is rarely seen in the world of FPGA designs." Your paper is however missing one existing solution for dynamic memory management in FPGAs, which is implementing memory allocation and deallocation based on Worst-Fit algorithm for FPGAs and is presented as "Low Latency Hardware-Accelerated Dynamic Memory Manager for Hard Real-Time and Mixed-Criticality Systems" and also as "Hardware Dynamic Memory Manager for Hard Real-Time Systems".

Issue #2, I still see in your paper is that despite you described insertion and deletion operations relatively well, I do not see a clear explanation of how search operation is designed/implemented. How do you perform search from linked list in O(1) time? Does not the search time depend on the length of the linked list? Even if you perform a key comparison in all nodes in parallel, the time needed to transfer the search command with the key to all nodes is probably not constant. Same applies to the time needed to transfer the result from the nodes to the output of the top-level module. It should depend on the number of nodes and the physical distance between all nodes and the output port of the module. So even if you use this approach of shared bus (also widely used in Shift Registers architecture), the latency depends on the number of nodes parameter. Please explain this in the paper (text-based description at least, ideally with a block diagram too).

Round 4

Reviewer 2 Report

Comments and Suggestions for Authors

All my previous comments were addressed in the paper.

I have no further objections for this paper.

I recommend to accept and publish this paper.